# Rethinking the Design Space of Reinforcement Learning for Diffusion Models: On the Importance of Likelihood Estimation Beyond Loss Design

**Jaemoo Choi** [1] [*]  **Yuchen Zhu** [1] [*]  **Wei Guo** [1]  **Petr Molodyk** [1]  **Bo Yuan** [1]  **Jinbin Bai** [2]  **Yi Xin** [3]
**Molei Tao** [1]  **Yongxin Chen** [1]

Code: https://github.com/jaemoo-choi/rethinking-diffusion-rl

## Abstract

Reinforcement learning has been widely applied to diffusion and flow models for visual tasks such as text-to-image generation. However, these tasks remain challenging because diffusion models have intractable likelihoods, which creates a barrier for directly applying popular policy-gradient type methods. Existing approaches primarily focus on crafting new objectives built on already heavily engineered LLM objectives, using ad hoc estimators for likelihood, without a thorough investigation into how such estimation affects overall algorithmic performance. In this work, we provide a systematic analysis of the RL design space by disentangling three factors: i) policy-gradient objectives, ii) likelihood estimators, and iii) rollout sampling schemes. We show that adopting an evidence lower bound (ELBO) based model likelihood estimator, computed only from the final generated sample, is the dominant factor enabling effective, efficient, and stable RL optimization, outweighing the impact of the specific policy-gradient loss functional. We validate our findings across multiple reward benchmarks using SD 3.5 Medium, and observe consistent trends across all tasks. Our method improves the GenEval score from $0.24$ to $0.95$ in 90 GPU hours, which is $4.6\times$ more efficient than FlowGRPO and $2\times$ more efficient than the SOTA method DiffusionNFT without reward hacking.

## 1. Introduction

Diffusion and flow models (Ho et al., 2020; Song et al., 2020; Liu et al., 2022; Lipman et al., 2022) have become a dominant paradigm for text-to-image (Esser et al., 2024; Labs, 2024) or text-to-video (Wan et al., 2025) synthesis, enabling strong generative performance through iterative denoising processes. There has been growing interest in post-training diffusion models (Black et al., 2023; Fan et al., 2023; Domingo-Enrich et al., 2024; Liu et al., 2025b; Zheng et al., 2025c), where external reward signals, such as human preferences (Wu et al., 2023; Xu et al., 2023; Hessel et al., 2021), or task-specific objectives (Ghosh et al., 2023) are used to guide generation toward desired outcomes. This line of work offers a flexible alternative to supervised fine-tuning (Zhang et al., 2023a) and has the potential to support fine-grained control and alignment without curated datasets.

Most attempts at reinforcement learning (RL) for diffusion models (Fan et al., 2023; Liu et al., 2025b) approach the problem by making minor modifications to PPO (Schulman et al., 2017) or GRPO (Shao et al., 2024) style policy-gradient objectives, with the ambition to replicate the tremendous, widely-seen success of RL in enhancing LLMs (Guo et al., 2025; Kimi Team et al., 2025) in visual tasks. However, using policy gradients for diffusion models is fundamentally challenging, due to the fact that policy gradient methods require exact, efficiently computable likelihoods (which fit autoregressive LLM naturally), yet diffusion models fail to provide as not being a likelihood-based generative model (Song et al., 2020; Benton et al., 2024).

FlowGRPO (Liu et al., 2025b), as the first successful practice in adapting GRPO to image generation tasks, fully inherits the loss objective of the GRPO and uses Gaussian transition from discretized reverse SDE sampling trajectories to estimate model likelihood. This naturally requires storing the entire sampling path; therefore, it is memory- and compute-intensive, leading to slow convergence. More recently, several works (Xue et al., 2025a; Zheng et al., 2025c) have shown that RL fine-tuning can instead be performed by operating only on the final generated sample, leading to

---

[1]Georgia Institute of Technology [2]National University of Singapore [3]Nanjing university. Correspondence to: Jaemoo Choi <jchoi843@gatech.edu>, Yongxin Chen <yongchen@gatech.edu>.

*Proceedings of the 43rd International Conference on Machine Learning*, Seoul, South Korea. PMLR 306, 2026. Copyright 2026 by the author(s).

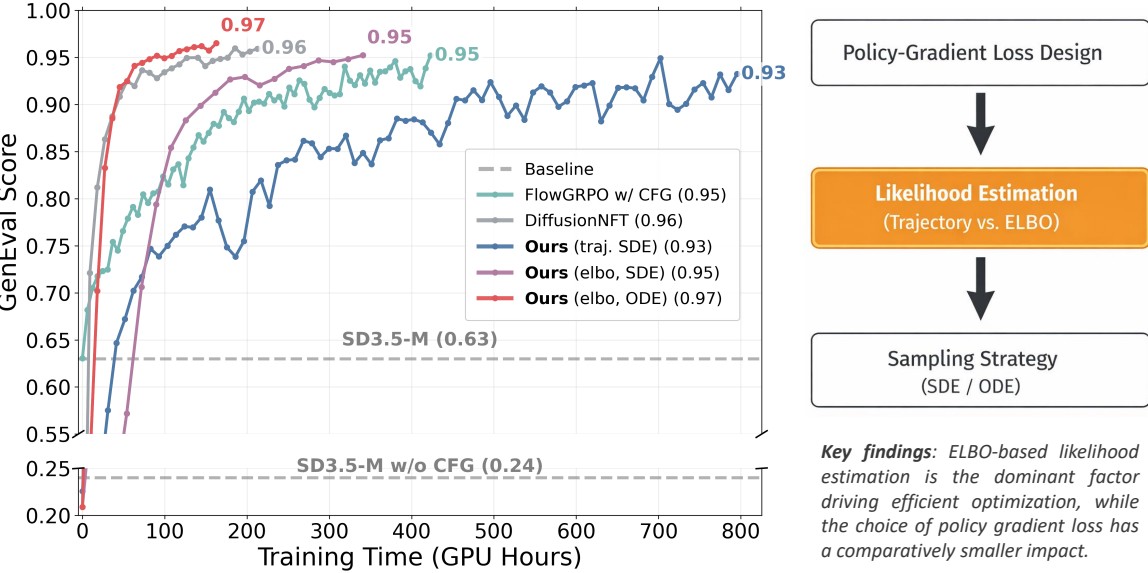

*Figure 1.* **Training efficiency and design-space analysis for reward-based diffusion fine-tuning.** *(Left)* GenEval performance across training time for various fine-tuning methods on SD3.5-Medium. *(Right)* Conceptual summary of the design space considered in this work, highlighting policy-gradient loss design, likelihood estimation, and sampling strategy.

substantial reductions in memory usage and improvements in computational efficiency. While these works have made important breakthroughs, their studies are largely empirical, and the mechanisms underlying their effectiveness have not been systematically analyzed. In particular, we seek answers to the following questions in this paper:

> *Which components in the RL design space, the policy gradient objectives, likelihood estimators, or sampling method, are primary drivers of efficiency and performance?*

To answer this question, we conduct a systematic study to disentangle the effects of three key design choices in RL-based diffusion fine-tuning: (i) the form of the policy-gradient objectives, (ii) the likelihood estimation recipe, and (iii) the choice of sampling strategy. For (i), we explore the standard GRPO, along with three new, theoretically grounded policy-gradient objectives that are lightweight in design and engineering-trick-free. For (ii), we conduct a principled investigation on various likelihood estimation approaches, including both backward trajectory-based estimators and forward ELBO-based estimators. For (iii), we compare the effects of the SDE-based and ODE-based samplers whenever a fair comparison is feasible to evaluate the impact of the inference scheme on the algorithm efficiency and stability. We design the controlled experiments to identify the core factors in the design space.

Through carefully designed numerical experiments on Stable Diffusion 3.5 Medium (SD3.5-M) (Esser et al., 2024), we observe that **the quality of likelihood estimators** is

central to RL of diffusion models. We note that the employment of **ELBO-based** likelihood approximation is the primary factor driving both optimization efficiency and performance gains, with a substantially greater impact than the specific choice of policy-gradient objective or sampler. We find that the trajectory-based estimator adopted by Flow-GRPO (Liu et al., 2025b) consistently causes a slow convergence and high computational cost, whereas ELBO-based estimators outperform it by achieving a better peak performance at a considerably faster rate, due to a decoupling between training and sampling dynamics that enables the use of any black-box solvers. The superior performance of ELBO-based likelihood estimation is observed across different policy-gradient objectives, indicating that algorithmic efficacy is largely determined by the likelihood estimation strategy rather than the loss objective itself. We further show that ODE-based sampling provides additional efficiency and stability benefits, as it requires a small number of function evaluations (as few as 10 steps) and matches the deterministic sampling procedure used at evaluation time.

By carefully studying each component of the design space, we also discovered a new method that efficiently achieves **state-of-the-art performance across multiple reward and benchmarks**, including GenEval (Ghosh et al., 2023), OCR (Liu et al., 2025b), and DrawBench (Ledig et al., 2017). Moreover, our discovered algorithm is up to $4.6\times$ more efficient than FlowGRPO (Liu et al., 2025b) and $2\times$ more efficient than the current SOTA method DiffusionNFT (Zheng et al., 2025c), which elevates GenEval score to from $0.24$ to

0.95 in less than 90 GPU hours. The fact that this success is achieved without introducing complex designs underscores the importance and the unlimited potential of likelihood estimation in advancing RL algorithms for diffusion and flow models.

## 2. Background

**Notation and Setup** Let $\mathcal{X}$ be a general state space (e.g., $\mathbb{R}^d$). Let the policy $\pi_\theta : \mathcal{X} \to \mathbb{R}$ be the output distribution of a likelihood-based generative model associated with parameters $\theta$, and $\pi_{\text{ref}}$ stands for a pretrained base model of the same type as $\pi_\theta$. The generation using these models is often conditioned on a fixed prompt $\boldsymbol{c}$, and hereafter, we assume that $\boldsymbol{c}$ is always included in the model input and omit it for simplicity. We use $\boldsymbol{x}^{1:G}$ to denote a group of $G$ responses generated given the same prompt $\boldsymbol{c}$, sg to denote the stop gradient operation. For simplicity, we also abbreviate the policy ratio as,

$$\rho_\theta(\boldsymbol{x}) = \frac{\pi_\theta(\boldsymbol{x})}{\pi_{\theta_{\text{old}}}(\boldsymbol{x})}, \; \text{sg}(\rho_\theta)(\boldsymbol{x}) = \frac{\text{sg}(\pi_\theta)(\boldsymbol{x})}{\pi_{\theta_{\text{old}}}(\boldsymbol{x})}$$

The task of generative models post-training through RL is to maximize a given reward function $R : \mathcal{X} \to \mathbb{R}$,

$$\max_\theta \; \mathop{\mathbb{E}}_{\boldsymbol{x} \sim \pi_\theta} \big[ R(\boldsymbol{x}) \big] - \beta \, \text{KL}(\pi_\theta || \pi_{\text{ref}}), \quad (1)$$

where $\beta$ controls the strength of the KL regularization to the base (pretrained) model, $\text{KL}(\pi_\theta || \pi_{\text{ref}}) = \mathbb{E}_{\pi_\theta} \log \frac{\pi_\theta}{\pi_{\text{ref}}}$ is the reverse KL between $\pi_\theta$ and $\pi_{\text{ref}}$. The optimal solution to problem (1) is $\pi_*(\boldsymbol{x}) \propto \pi_{\text{ref}}(\boldsymbol{x}) \exp(R(\boldsymbol{x})/\beta)$. The algorithms for solving such a problem have been extensively studied in the RL literature of likelihood-based generative models, particularly RL post-training for LLMs. The existing effective approaches are policy-gradient-based methods, such as KL-regularized REINFORCE (Sutton et al., 1999; Li & He, 2025) and its variants, including PPO (Schulman et al., 2017) and GRPO (Shao et al., 2024).

**Policy Gradient Methods** KL-regularized REINFORCE considers,

$$\mathcal{L}_{\text{reinfo}}(\theta) = \mathop{\mathbb{E}}_{\boldsymbol{x}^i \sim \pi_{\theta_{\text{old}}}} \Big[ \rho_\theta(\boldsymbol{x}^i) A^i - \beta \, \text{kl}(\boldsymbol{x}^i) \Big]; \quad (2)$$

GRPO considers a more complicated objective,

$$\mathcal{L}_{\text{grpo}}(\theta) = \mathop{\mathbb{E}}_{\boldsymbol{x}^i \sim \pi_{\theta_{\text{old}}}} \Big[ \min \Big( \rho_\theta(\boldsymbol{x}^i) A^i,$$
$$\text{clip}\big( \rho_\theta(\boldsymbol{x}^i), 1 - \varepsilon, 1 + \varepsilon \big) A^i \Big) - \beta \, \text{kl}(\boldsymbol{x}^i) \Big], \; \text{(GRPO)}$$

where $\varepsilon$ controls the strength of the clipping operations, and $\text{kl}(\boldsymbol{x}^i)$ is a per-sample KL estimator. For the purpose of lightweight and efficient training, the advantages are typically estimated from rewards within the output group:

$$A^i = \frac{R(\boldsymbol{x}^i) - \text{mean}(R(\boldsymbol{x}^1), \dots, R(\boldsymbol{x}^G))}{\text{std}(R(\boldsymbol{x}^1), \dots, R(\boldsymbol{x}^G))}$$

### 2.1. Diffusion and Flow Models

Diffusion and flow models (Song et al., 2020; Lipman et al., 2023) model continuous data distribution on $\mathbb{R}^d$ by gradually corrupting clean data $\boldsymbol{x}_0 \sim \pi_0 = p_{\text{data}}$ with additive Gaussian noise according to a forward process, while generation is achieved through reversing this process. The forward-noising process with noise schedule $\alpha_t, \sigma_t$ is

$$\text{d}\boldsymbol{x}_t = \frac{\dot{\alpha}_t}{\alpha_t} \boldsymbol{x}_t \text{d}t + \sqrt{2\dot{\sigma}_t \sigma_t - 2\frac{\dot{\alpha}_t}{\alpha_t}\sigma_t^2} \text{d}\boldsymbol{w}_t, \quad (\overrightarrow{\mathbb{P}})$$

where $\dot{\alpha}_t$ and $\dot{\sigma}_t$ are time derivatives, $\boldsymbol{w}_t$ is standard Brownian motion, and ($\overrightarrow{\mathbb{P}}$) admits a closed-form expression:

$$\boldsymbol{x}_t \sim p_{t|0}(\boldsymbol{x}_t | \boldsymbol{x}_0) \iff \boldsymbol{x}_t = \alpha_t \boldsymbol{x}_0 + \sigma_t \boldsymbol{\epsilon}, \boldsymbol{\epsilon} \sim \mathcal{N}(\boldsymbol{0}, \boldsymbol{I}).$$

Diffusion and flow models can be represented through the velocity parameterization $\boldsymbol{v}_\theta$ and trained through minimizing the evidence lower bound (ELBO) of data $\boldsymbol{x}_0$,

$$\text{ELBO}(\boldsymbol{v}_\theta, \boldsymbol{x}_0) = \mathop{\mathbb{E}}_{\boldsymbol{x}_t \sim p_{t|0}(\cdot|\boldsymbol{x}_0)} [w(t) \| \boldsymbol{v}_\theta(\boldsymbol{x}_t, t) - \boldsymbol{v} \|_2^2],$$

where $\boldsymbol{v} = \dot{\alpha}_t \boldsymbol{x}_0 + \dot{\sigma}_t \boldsymbol{\epsilon}$ is the tangent of the conditional trajectory, $w(t)$ is a weighting function. Flow models typically refer to a special case of the above setting with $\alpha_t = 1 - t$, $\sigma_t = t$, and thus $\boldsymbol{v} = \boldsymbol{\epsilon} - \boldsymbol{x}_0$. We stick to this setup in the sequel for simplicity. With the learned velocity $\boldsymbol{v}_\theta$, the sampling follows the reverse dynamics,

$$\text{d}\boldsymbol{x}_t = \big[ \boldsymbol{v}_\theta(\boldsymbol{x}_t, t) + \frac{g_t^2}{2t} \big( \boldsymbol{x}_t + (1 - t)\boldsymbol{v}_\theta(\boldsymbol{x}) \big) \big] \text{d}t + g_t \text{d}\boldsymbol{w}_t,$$

$$g_t = a\sqrt{\frac{2t}{1 - t}}, \quad a \in [0, 1]. \quad (\overleftarrow{\mathbb{P}})$$

Here, $a$ controls the level of stochasticity of the trajectories.

## 3. Rethinking RL for Diffusion Model with Policy-Gradient Methods

Existing RL approaches for diffusion models, such as Flow-GRPO and its variants (Liu et al., 2025b; He et al., 2025; Wang et al., 2025a), predominantly aim to directly adapt policy-gradient methods from the LLM-RL literature, such as GRPO, to score-based generative models. However, GRPO-type objectives are heavily engineered, with many complex tricks that may be tailored only to LLMs. Because diffusion and flow models are, by design, not likelihood-based generative models (Song et al., 2020; Benton et al., 2024), we need a thorough re-evaluation of the policy gradient objectives used in this task, dissecting each component to distinguish between non-essential tricks and designs and core contributors to algorithmic success.

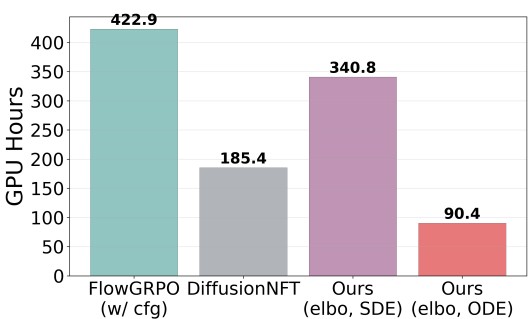

*Figure 2.* **Training time comparison on GenEval.** We report the total GPU hours (8×H100) required to reach a GenEval score of 0.95 for different fine-tuning methods. ELBO-based likelihood estimation substantially reduces training cost compared to trajectory-based approaches, and ODE sampling further improves efficiency while achieving the same target performance.

### 3.1. Revisiting the Vanilla Policy Gradient Method

We recall the vanilla **exact policy gradient (EPG)** objective for solving the reward alignment task (1):

$$\mathcal{L}_{\mathrm{epg}}(\theta) = \mathbb{E}_{\boldsymbol{x}_0^i \sim \pi_{\theta_{\mathrm{old}}}} \Big[ \mathrm{sg}(\rho_\theta)(\boldsymbol{x}_0^i) A_{\mathrm{epg}}^i \log \pi_\theta(\boldsymbol{x}_0^i)$$
$$- \beta \, \mathrm{kl}(\boldsymbol{x}_0^i) \Big], \ A_{\mathrm{epg}}^i = R^i - \mathrm{mean}(R^1, \dots, R^G). \quad \text{(EPG)}$$

EPG is essentially the same as KL-regularized REINFORCE with an advantage function computed by subtracting the group mean from each reward value.[1] Compared with the GRPO-based diffusion RL approach, EPG follows three simplifications:

**Remove Clipping Operation** (EPG) completely discards the use of $\mathrm{clip}(\cdot, 1 - \varepsilon, 1 + \varepsilon)$, in which the clipping threshold $\varepsilon$ is an important yet notoriously hard-to-tune hyperparameter, which even necessitates asymmetric thresholds in certain cases (Chen et al., 2025a; Khatri et al., 2025). With the clipping operation present, the clipping threshold needs to stay at a moderate level, without being too large, which leads to potential training instability, or too small, which hinders training efficiency and effectiveness. This complex dilemma, compounded by additional uncertainty in likelihood estimation from diffusion models, leaves it unclear whether it's necessary to use the clipping operation. To keep it simple, we avoid using $\mathrm{clip}$ in the objective formulation and find that it has a negligible impact on final performance.

**Remove Advantage Bias** (EPG) drops the advantage normalization by dividing the standard deviation computed within the response group. Such normalization is known to cause prompt-level difficulty bias in LLM-based RL for

0/1-verifiable rewards, where tasks that are too easy or too difficult are often associated with lower reward standard deviations, thereby biasing RL updates toward these cases (Liu et al., 2025c). This issue is exacerbated in diffusion model RL, where reward values are often distributed continuously, with even fewer discrepancies. To ensure unbiased training and a better numerical stability, we avoid dividing centered reward values by their standard deviation.

**Remove Guided Generation** (EPG) adopts the naive conditional sampling from the diffusion/flow models without using classifier-free guidance (CFG) (Ho & Salimans, 2022), which is adopted as a default option by most of the diffusion model RL literature. While CFG substantially improves output quality, it also doubles the sampling cost, making training more computationally intensive. More importantly, CFG potentially causes a training-inference distribution mismatch, as the guided output distribution is known to be sharper and more interior-concentrated than the non-guided one, which we aim to estimate and incorporate into the objective. Such a mismatch has been shown to cause issues for policy-gradient objectives in LLM RL training (Zheng et al., 2025b; Liu et al., 2025a). To take preventive measures, we eliminate the use of CFG in rollout sampling, thereby avoiding the distribution mismatch while achieving an additional training speedup by saving NFEs.

### 3.2. Keystone in Diffusion RL: Likelihood Estimation

**Data Likelihood Estimation** Unlike LLM, diffusion and flow models do not have direct access to data likelihood, which is essential to apply policy-gradient-based RL techniques for post-training. The recipe for estimating data likelihood has three main ingredients: the estimation formula, the choice of sampler, and the ELBO weighting. We will elaborate on each of these aspects in this section.

**Estimation Formula** There are two major ways of obtaining the likelihood of diffusion-generated data, using either the forward process ($\overrightarrow{\mathbb{P}}$) or the backward process ($\overleftarrow{\mathbb{P}}$).

Using the **forward process** ($\overrightarrow{\mathbb{P}}$), the likelihood $\pi_\theta(\boldsymbol{x}_0)$ can be accurately approximated up to a training-unimportant constant $\boldsymbol{C}_{\mathrm{fw}}(\boldsymbol{x}_0)$, using the flow matching loss (Song et al., 2021b; Kingma et al., 2021; Kingma & Gao, 2023),[2]

$$\log \pi_\theta(\boldsymbol{x}_0) = -\mathbb{E}_{t,\boldsymbol{\epsilon}} \big[ w(t) \| \boldsymbol{v}_\theta(\boldsymbol{x}_t, t) - \boldsymbol{v} \|_2^2 \big]. \quad \text{(ELBO)}$$

Using the **backward process** ($\overleftarrow{\mathbb{P}}$), note that the transition probability becomes a tractable Gaussian distribution,

$$p_\theta(\boldsymbol{x}_{t_{i-1}} | \boldsymbol{x}_{t_i}) = \mathcal{N}\Big( \boldsymbol{x}_{t_i} + \Big[ \boldsymbol{v}_\theta(\boldsymbol{x}_{t_i}, t_i) + \frac{g_{t_i}^2}{2t_i}\big(\boldsymbol{x}_{t_i} +$$
$$(1 - t_i)\boldsymbol{v}_\theta(\boldsymbol{x}_{t_i}, t_i)\big) \Big] (t_i - t_{i-1}), g_{t_i}^2(t_i - t_{i-1})\boldsymbol{I} \Big).$$

---

[1]EPG differs from Peng et al. (2019): EPG is an on-policy method that corrects for distributional shift via the importance ratio $\mathrm{sg}(\rho_\theta)$, whereas AWR is an off-policy weighted regression approach that does not employ importance ratio correction.

[2]The standard flow matching loss corresponds to a special case of (ELBO) with uniform weighting $w(t) = 1$ (cf. (14)).

| SD3.5-M (w/o CFG) | SD3.5-M (w/ CFG) | FlowGRPO | AWM (w/o CFG) | DiffusionNFT | Ours (elbo, ODE) |
|---|---|---|---|---|---|

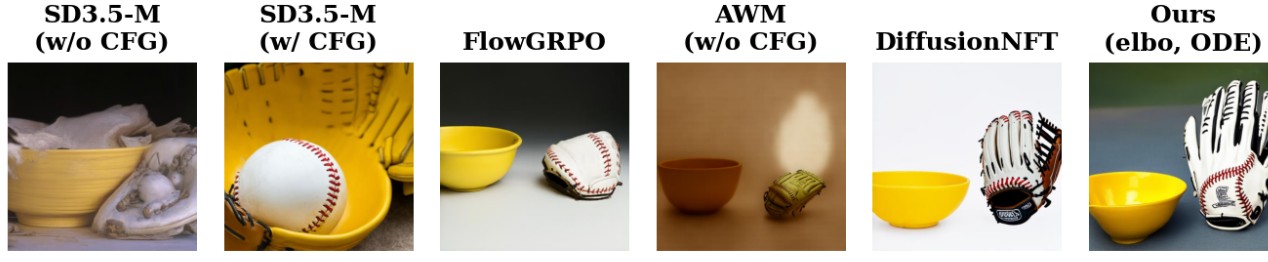

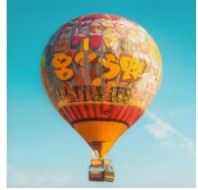 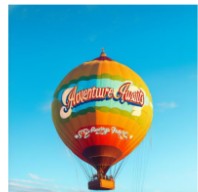 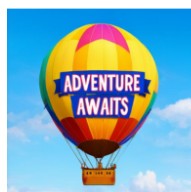 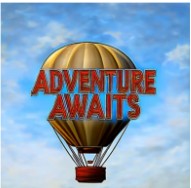 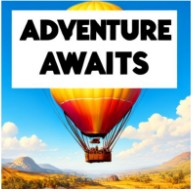 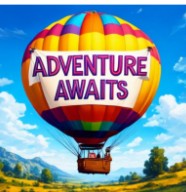

a photo of a yellow bowl and a white baseball glove

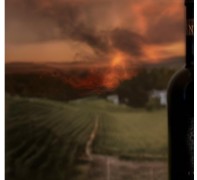 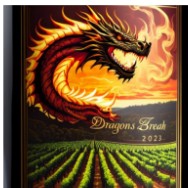 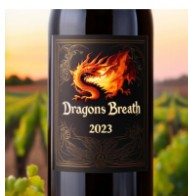 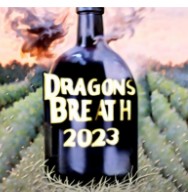 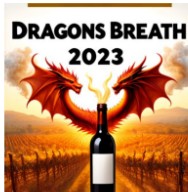 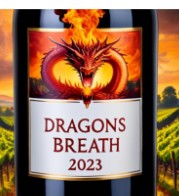

A vibrant hot air balloon ascends into a clear blue sky, trailing a banner that reads "Adventure Awaits" in bold, flowing letters. The balloon's colorful pattern contrasts beautifully against the serene landscape

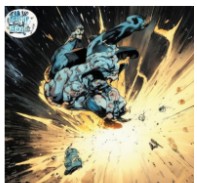 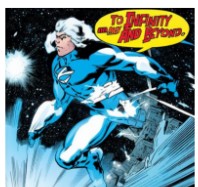 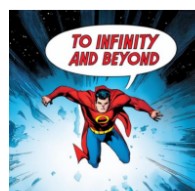 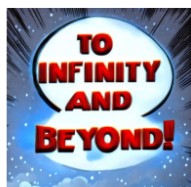 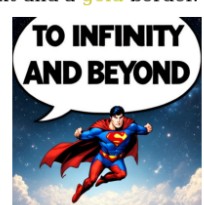 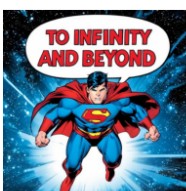

A close-up of a wine bottle with an intricate label titled "Dragons Breath 2023", featuring a fiery dragon exhaling smoke over a vineyard at sunset, with elegant cursive text and a gold border.

A vibrant comic book panel featuring a dynamic superhero leaping into the sky, with a bold speech bubble reading "To Infinity And Beyond" floating above their head, set against a starry backdrop.

*Figure 3.* Qualitative comparison between benchmarks and our model. See App. D for additional figures.

We can derive a different estimator for $\pi_\theta(\boldsymbol{x}_0)$ up to a training-unimportant constant $\boldsymbol{C}_{\text{bw}}(\boldsymbol{x}_0)$ (Ho et al., 2020),

$$\log \pi_\theta(\boldsymbol{x}_0) \approx \sum_{i=1}^{N} \log \pi_\theta(\boldsymbol{x}_{t_{i-1}}|\boldsymbol{x}_{t_i}) \qquad \text{(Trajectory)}$$

where $0 = t_0 < \cdots < t_N = 1$, $g_t = \sqrt{2t/(1-t)}$ to ensure $\boldsymbol{C}_{\text{bw}}(\boldsymbol{x}_0)$ is a $\theta$-independent constant.

**Sampler Choice** The availability of the two estimation formulas is dependent on the choice of the sampler. The mainstream inference methods are SDE-based sampling (Ho et al., 2020), which corresponds to using $a = 1$ in ($\overleftarrow{\mathbb{P}}$), and ODE-based sampling (Song et al., 2021a), which corresponds to using $a = 0$. For fast and NFE-efficient sampling, the ODE sampler is more preferable due to the many successes of inference acceleration algorithms (Lu et al., 2022; Zhang & Chen, 2022; Zhang et al., 2023b).

The stochasticity of trajectories has an important influence on the validity of estimation formulas. Trajectory-based formula (Trajectory) can only be computed with the SDE sampler, as otherwise the estimation formula would be problematic due to the degeneracy of the conditional Gaussian transition $p_\theta(\boldsymbol{x}_{t_{i-1}}|\boldsymbol{x}_{t_i})$. This constrains trajectory-based likelihood estimation to a high computational cost.

On the other hand, ELBO-based likelihood estimation (ELBO) is free of such constraints, and can be deployed with any black-box sampler. Moreover, ELBO computation does not require inference trajectories; therefore, we can cache only the final generated samples and discard the intermediate diffusion states, saving additional memory. This difference makes the ELBO-based approach more flexible and efficient than the trajectory one.

**ELBO Weighting** While the trajectory-based estimation

has a fixed weighting implicitly determined by the noise schedule $g_t$, ELBO-based estimation enjoys an additional flexibility, namely the weighting $w(t)$. The ELBO estimation variants differ mostly in how they choose $w(t)$ and the regression objective (i.e., the $\epsilon, \boldsymbol{x}, \boldsymbol{v}$-loss) (Li & He, 2025). For the simplicity of demonstration, we unified each objective considered in this work in the same form of $\boldsymbol{v}$-loss $\mathbb{E}\|\boldsymbol{v}_\theta - \boldsymbol{v}\|_2^2$ with different weighting. We consider several choices in the literature, including **Path-KL weighting**, which amounts to picking $w(t) = \frac{1-t}{t}$ (Song et al., 2020), and **Simple weighting**, which is equivalent to choosing $w(t) = 1$ (Ho et al., 2020; Shi & Titsias, 2025). Additionally, we investigate a self-normalized data-dependent **Adaptive weighting** (Yin et al., 2024; Zheng et al., 2025c), which computes the ELBO as $\mathbb{E}_{t,\epsilon}\left[t \cdot d \cdot \|\boldsymbol{v}_\theta - \boldsymbol{v}\|_2^2 / \text{sg}(\|\boldsymbol{v}_\theta - \boldsymbol{v}\|_1)\right]$. For a detailed discussion, see App. B.1.

### 3.3. Alternatives of Policy-Gradient Objectives

With likelihood estimation as the primary focus in diffusion model RL, the choice of policy-gradient objectives can be relatively flexible once an effective likelihood estimation method is adopted. To illustrate this point, we can consider a few objectives that differ from the EPG objective introduced in Sec. 3.1. When combined with a high-quality likelihood estimator, these objectives exhibit near-identical peak performance without using CFG during training.

**GRPO** The first objective candidate to consider is GRPO, which is described in (GRPO). We retain all the original design elements, including the clipping operation and the original advantage normalization method. We adapt this objective to the diffusion and flow model by estimating $\pi_\theta$ using the adaptive weighted ELBO (15) with a *single* Monte Carlo sample, a recipe that consistently performs well across tasks. This computes $\pi_\theta(\boldsymbol{x}) / \pi_{\theta_{\text{old}}}(\boldsymbol{x})$ as

$$\exp\left(\frac{td \cdot \|\boldsymbol{v}_{\text{old}}(\boldsymbol{x}_t, t) - \boldsymbol{v}\|_2^2}{\text{sg}(\|\boldsymbol{v}_{\text{old}}(\boldsymbol{x}_t, t) - \boldsymbol{v}\|_1)} - \frac{td \cdot \|\boldsymbol{v}_\theta(\boldsymbol{x}_t, t) - \boldsymbol{v}\|_2^2}{\text{sg}(\|\boldsymbol{v}_\theta(\boldsymbol{x}_t, t) - \boldsymbol{v}\|_1)}\right).$$

We also include a discussion of how other GRPO-style algorithms can be connected to our approach in App. B.2.

**Proximal Exact Policy Gradient (PEPG)** To ensure stable policy optimization, one may constrain policy updates to remain close to the current policy, as in trust-region methods such as TRPO (Schulman et al., 2015). PPO (Schulman et al., 2017) relaxes these constraints using heuristic tricks such as clipping. Since we have completely abandoned the use of clip, we propose to achieve a similar stabilization goal through a theoretically-grounded lens of proximal gradient descent over the space of probability measures. Concretely,

we consider the following optimization problem:

$$\max_{\pi_\theta} \quad \mathbb{E}_{\boldsymbol{x} \sim \pi_{\theta_{\text{old}}}}\left[\frac{\pi_\theta(\boldsymbol{x})}{\pi_{\theta_{\text{old}}}(\boldsymbol{x})} R(\boldsymbol{x})\right] - \beta \, \text{KL}(\pi_\theta \| \pi_{\text{ref}})$$
$$- \frac{1}{\eta} \text{KL}(\pi_\theta \| \pi_{\theta_{\text{old}}}),$$

where the first KL term regularizes toward the reference model and the second KL term acts as a proximal penalty that limits the step size relative to the old policy. Solving this problem yields the novel PEPG objective,

$$\mathcal{L}_{\text{pepg}}(\theta) = \mathbb{E}_{\boldsymbol{x}_0^i \sim \pi_{\theta_{\text{old}}}}\left[\left(A_{\text{pepg}}^i - \log \text{sg}(\rho_\theta)(\boldsymbol{x}_0^i)\right)\rho_\theta(\boldsymbol{x}_0^i)\right.$$
$$\left. - \eta \, \text{kl}(\boldsymbol{x}_0^i)\right], \qquad A_{\text{pepg}}^i = \frac{\eta}{\beta} A_{\text{epg}}^i, \quad \text{(PEPG)}$$

where $\eta$ is another hyperparameter different from $\beta$ that controls the step size for such proximal gradient descent over probability distribution space.

**Proximal Advantage Regression (PAR)** Alternatively, we can achieve the same goal targeted by PEPG through an $L^2$ regression-type loss between the log probability ratio and the advantage value:

$$\mathcal{L}_{\text{par}}(\theta) = \mathbb{E}_{\boldsymbol{x}_0^i \sim \pi_{\theta_{\text{old}}}}\left[-\frac{1}{2} \text{sg}(\rho_\theta)(\boldsymbol{x}_0^i)\left\| A_{\text{par}}^i - \log \rho_\theta(\boldsymbol{x}_0^i)\right\|_2^2\right.$$
$$\left. - \eta \, \text{kl}(\boldsymbol{x}_0^i)\right], \qquad A_{\text{par}}^i = \frac{\eta}{\beta} A_{\text{epg}}^i. \quad \text{(PAR)}$$

While our proposed EPG, PEPG, and PAR share distinct loss formulations, they are all mathematically valid policy gradient objectives that provably achieve the goal of solving post-training task (1), as is stated in Thm. 3.1. We provide proof in App. A.

**Theorem 3.1** (Mathematical Validity of PG Objectives). (EPG), (PEPG), *and* (PAR) *share the optimal minimizer* $\pi_*(\boldsymbol{x}) \propto \pi_{ref}(\boldsymbol{x}) \exp(R(\boldsymbol{x})/\beta)$.

**General algorithm** Alg. 1 summarizes a unified training procedure. It highlights three interchangeable components: (i) the policy-gradient objective (EPG, PEPG, PAR, GRPO, etc.), (ii) the likelihood estimator (trajectory-based or ELBO-based), and (iii) the sampler (SDE or ODE). When ELBO-based estimation is used, the update depends only on final samples, enabling the use of deterministic ODE sampling during training.

## 4. Experiments

Our experiments are designed to analyze the key design choices in reward-based diffusion fine-tuning and to identify the factors that most strongly influence optimization efficiency and performance. In Sec. 4.2, we study the impact

*Table 1.* **Effect of likelihood estimation and sampling strategy across policy-gradient objectives on `GenEval`.** Gray-colored cells indicate in-domain reward. Across objectives, we compare trajectory-based (w/ SDE sampling) and ELBO-based (w/ SDE or ODE) likelihood estimation. Performance differences across policy-gradient objectives are minor, indicating limited sensitivity to the specific loss formulation. Under ELBO estimation, ODE sampling achieves performance comparable to SDE sampling with reduced training cost.

| Loss | Likelihood Est. | Sampler | GenEval | PickScore | ClipScore | HPSv2.1 | Aesthetic | ImgRwd |
|---|---|---|---|---|---|---|---|---|
| (EPG) | Traj. | SDE | 0.92 | 21.39 | 0.301 | 0.240 | 4.55 | 0.83 |
| | Traj. | SDE w/ cfg | 0.95 | 22.48 | 0.308 | 0.263 | 5.12 | 1.15 |
| | ELBO | SDE | 0.90 | 22.00 | 0.297 | 0.252 | 5.03 | 0.75 |
| | ELBO | ODE | 0.96 | 22.77 | 0.304 | 0.281 | 5.33 | 1.19 |
| (PEPG) | ELBO | SDE | 0.96 | 23.25 | 0.305 | 0.302 | 5.47 | 1.35 |
| | ELBO | ODE | 0.96 | 22.85 | 0.305 | 0.289 | 5.33 | 1.26 |
| (PAR) | ELBO | SDE | 0.94 | 22.79 | 0.300 | 0.281 | 5.26 | 1.16 |
| | ELBO | ODE | 0.96 | 22.97 | 0.302 | 0.300 | 5.42 | 1.35 |
| (GRPO) | ELBO | ODE | 0.94 | 22.45 | 0.306 | 0.272 | 5.10 | 1.03 |

of **(i)** policy-gradient loss design, **(ii)** likelihood estimation strategy, and **(iii)** sampling scheme. In Sec. 4.3, we provide additional empirical analysis and ablation studies.

### 4.1. Experimental Setup

All experiments are conducted using `SD3.5-M` (Esser et al., 2024) at a resolution of $512 \times 512$. Unless otherwise stated, our training configuration follows DiffusionNFT (Zheng et al., 2025c) to ensure a fair comparison. We fine-tune the pretrained diffusion model using LoRA (Hu et al., 2022). We refer readers to App. C for detailed experimental settings.

**Reward Models** We consider both rule-based and model-based rewards. Rule-based rewards include `GenEval` (Ghosh et al., 2023) for compositional image generation and `OCR` for visual text rendering, following the partial reward assignment strategies used in FlowGRPO. Model-based rewards include `PickScore` (Kirstain et al., 2023), `CLIPScore` (Hessel et al., 2021), `HPSv2.1` (Wu et al., 2023), `Aesthetics` (Schuhmann, 2022), `ImageReward` (Xu et al., 2023), which capture image quality, image–text alignment, and human preference.

**Prompt Datasets** For `GenEval` and `OCR`, we use the corresponding training and test splits provided by FlowGRPO. For other reward models, training is performed on `Pick-a-Pic` (Kirstain et al., 2023), and evaluation is conducted on `DrawBench` (Saharia et al., 2022).

### 4.2. Main Results

**Policy-gradient loss design has a limited impact** As shown in Tab. 1, we observe comparable `GenEval` performance across different policy-gradient objectives, including (GRPO), (EPG), (PEPG), and (PAR). This suggests that, once likelihood estimation and sampling strategy are fixed, the specific choice of policy-gradient loss functional has a relatively minor effect on final performance.

Although performance across different policy-gradient losses is largely comparable, (PEPG) and (PAR) achieve slightly higher average performance than others, and they both correspond to an *exact* proximal policy-gradient formulation developed in this work. For other benchmarks, unless otherwise specified, *we select PEPG as the primary objective for evaluation*.

**ELBO-based likelihood estimation accelerates convergence** Fig. 4 compares convergence behavior across different likelihood estimation strategies in terms of prompt efficiency. ELBO-based likelihood estimation reaches a `GenEval` score of 0.95 after observing substantially fewer training prompts than trajectory-based methods. In particular, ELBO with ODE sampling achieves approximately $4.68\times$ faster convergence, while ELBO with SDE sampling achieves $1.24\times$ faster convergence relative to FlowGRPO. These results indicate that ELBO-based likelihood estimation provides a significantly more efficient optimization signal, enabling rapid performance gains with reduced training data and computation.

**ODE sampling further boosts efficiency under ELBO** Given ELBO-based likelihood estimation, the choice of sampler primarily affects computational efficiency over performance. As shown in Fig. 4, ODE- and SDE-based samplers observe a similar number of training prompts to reach comparable `GenEval` performance. However, SDE sampling incurs substantially higher computational cost, as it requires approximately 4 times more function evaluations to generate stable, image-like samples (e.g., 40 steps v.s. 10 steps for ODE sampling). As a result, ODE sampling yields significantly faster training while maintaining comparable

*Table 2.* **Evaluation Results across tasks and reward settings.** Gray-colored cells indicate in-domain reward performance for each task. † indicates evaluated results on official checkpoints. ‡ indicates evaluated under $1024 \times 1024$ resolution. **Bold** indicates the best result within each task block. For models in Baselines category, we evaluate GenEval and OCR on its corresponding dataset, while other rewards are evaluated on DrawBench (Saharia et al., 2022).

| Eval. Dataset | Model | GenEval | OCR | PickScore | ClipScore | HPSv2.1 | Aesthetic | ImgRwd |
|---|---|---|---|---|---|---|---|---|
| Baselines | SD-XL‡ | 0.55 | 0.14 | 22.42 | 0.287 | 0.280 | 5.60 | 0.76 |
| | SD3.5-L‡ | 0.71 | 0.68 | 22.91 | 0.289 | 0.288 | 5.50 | 0.96 |
| | FLUX.1-Dev | 0.66 | 0.59 | 22.84 | 0.295 | 0.274 | 5.71 | 0.96 |
| | SD3.5-M | 0.24 | 0.12 | 20.51 | 0.237 | 0.204 | 5.13 | -0.58 |
| | + CFG | 0.63 | 0.59 | 22.34 | 0.285 | 0.279 | 5.36 | 0.85 |
| GenEval | FlowGRPO† | 0.95 | - | 22.51 | 0.293 | 0.274 | 5.32 | 1.06 |
| | AWM | 0.89 | - | 22.00 | 0.302 | 0.242 | 4.94 | 0.84 |
| | DiffusionNFT | 0.95 | - | **22.88** | 0.303 | **0.289** | 5.25 | 1.21 |
| | Ours | **0.96** | - | 22.85 | **0.305** | **0.289** | **5.33** | **1.26** |
| OCR | FlowGRPO† | - | 0.92 | 22.41 | 0.290 | 0.280 | 5.32 | 0.95 |
| | AWM | - | 0.80 | 20.70 | 0.301 | 0.206 | 4.53 | -0.13 |
| | DiffusionNFT | - | 0.93 | 22.09 | 0.307 | 0.277 | 5.17 | 0.97 |
| | Ours | - | **0.94** | **22.93** | **0.315** | **0.302** | **5.33** | **1.34** |
| Drawbench | FlowGRPO† | - | - | 23.50 | 0.280 | 0.316 | 5.90 | 1.29 |
| | DiffusionNFT | - | - | 23.61 | 0.288 | **0.344** | 6.04 | **1.46** |
| | Ours | - | - | **23.68** | **0.296** | 0.325 | **6.06** | 1.45 |

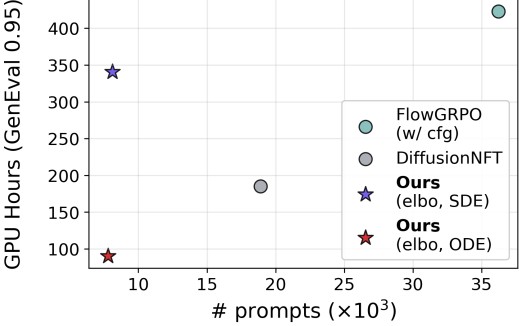

*Figure 4.* The **# of prompts** seen in training vs. **GPU hours** to reach `GenEval` score of 0.95.

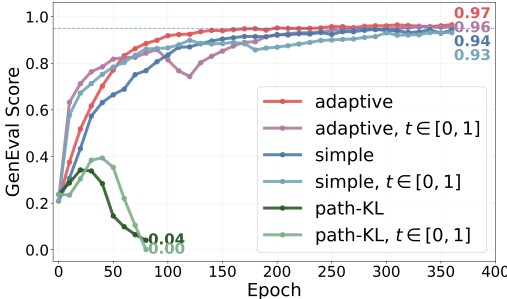

*Figure 5.* **Ablation on ELBO estimators** on `GenEval`.

performance under ELBO-based optimization.

**Comparison across benchmarks** In Tab. 2, we compare our method with PEPG, ELBO-based likelihood estimation, and ODE sampling against various recent benchmarks, including FlowGRPO (Liu et al., 2025b), AWM (Xue et al., 2025a), and DiffusionNFT (Zheng et al., 2025c), across multiple reward functions. Our approach outperforms existing methods in most cases while benefiting from improved training efficiency and a unified training procedure, suggesting a high efficacy of our algorithm recipe.

### 4.3. Further Discussion

**Ablation on ELBO Estimations** We study the effect of different ELBO estimation strategies. Specifically, we consider three ELBO formulations: a path-KL weighted estimator

(13), a simple weighted ELBO estimator (14), and an adaptive ELBO estimator (15). Each ELBO can be estimated using different Monte Carlo schemes, either by sampling a single diffusion timestep or by aggregating estimates across multiple timesteps over the entire diffusion trajectory. We evaluate all combinations using the PEPG objective with ELBO-based likelihood estimation and ODE sampling. As shown in Fig. 5, four ELBO estimators, except the two path-KL weighted ones, show stable training and strong performance. Moreover, both single-timestep and whole-timestep (i.e. $t \in [0, 1]$) estimation strategies achieve comparable results. Given its lower computational cost and simpler implementation, we recommend the single-timestep ELBO estimator in practice. Details for ELBO estimation are discussed in App. B.3.

**Clipping is not an important design choice** As shown in

Tab. 1, the GRPO and EPG objectives differ primarily in the use of clipping and normalization by the standard deviation. Removing these design choices yields comparable performance across benchmarks, indicating that such heuristics are not critical to achieving strong results in our setting. To further substantiate this claim, we note that GRPO with clipping under ELBO+ODE achieves a GenEval score of $0.94$, while EPG, PEPG, and PAR (all without clipping) achieve scores of approximately $0.96$, demonstrating that clipping offers no performance advantage and may even slightly hinder optimization. Similarly, advantage normalization by the standard deviation introduces a prompt-level difficulty bias: prompts that are too easy or too difficult tend to have smaller reward standard deviations, skewing updates toward these cases (Liu et al., 2025c). This effect is exacerbated for continuously distributed rewards common in diffusion model RL.

**Fine-tuning without CFG** Consistent with our formulation and supported by prior work (Zheng et al., 2025c), we train reward-based diffusion fine-tuning without CFG (Ho & Salimans, 2022). Our results show that competitive performance can be achieved even without CFG across benchmarks. Moreover, when combined with ELBO-based likelihood estimation and ODE sampling, CFG-free training yields substantial efficiency gains. Eliminating CFG further reduces the number of function evaluations required during sampling, simplifying both training and inference. Specifically, removing CFG reduces the number of function evaluations by approximately $2\times$ during sampling, yielding a substantial reduction in training cost without sacrificing performance. For these reasons, we recommend CFG-free fine-tuning as a practical and effective default.

## 5. Related Works

**RL for language models.** RL has been extensively verified as a principled method to enhance the model capability (Ouyang et al., 2022; Guo et al., 2025), where popular approaches include PPO (Schulman et al., 2017), GRPO (Shao et al., 2024) and its variants (Liu et al., 2025c; Yu et al., 2025; Ahmadian et al., 2024). Recent algorithmic developments on RL methods focus on new objective designs (Zhao et al., 2025; Zheng et al., 2025b; Gao et al., 2025; Chen et al., 2025a; Kimi Team et al., 2025) and their unbiased estimation (Zhang et al., 2025; Zheng et al., 2025a; Liu et al., 2025a) to ensure stable training.

**RL for Diffusion and Flow models.** RL has also been widely adopted to post-train diffusion and flow models to align model output with human preference (Fan et al., 2023; Black et al., 2023; Domingo-Enrich et al., 2024). Flow-GRPO (Liu et al., 2025b) first adapts GRPO to diffusion models through a rough trajectory-based likelihood estima-

tion for data likelihood and achieves decent results on text-to-image generation, followed by a series of works improving it (He et al., 2025; Wang et al., 2025a; Xue et al., 2025b; Wang et al., 2025b; Ye et al., 2025; Xue et al., 2025a). On a separate line of work, DiffusionNFT (Zheng et al., 2025c) adapts negative-finetuning (NFT) (Chen et al., 2025b) to diffusion and flow models. DPPO (Ren et al., 2025) applies PPO to diffusion policies for continuous control and robotic manipulation tasks, demonstrating that careful adaptation of policy-gradient methods to the diffusion framework can yield strong performance. Q-score matching (Psenka et al., 2024) takes a different approach by learning a diffusion model policy from reward signals via matching the score function with a Q-function gradient, bypassing explicit likelihood computation. Concurrently, our PEPG loss is closely related to QUATRO (Lee et al., 2026). The main difference is that QUATRO optimizes an exact trust-region bound objective, whereas our method adopts a proximal objective without considering any trust-region bounds.

## 6. Conclusion

In this work, we conducted a systematic study of the design space of RL for diffusion models. Through experiments on text-to-image generation tasks, we found that a high-quality likelihood estimator plays a more important role in algorithmic success than the specific choice of policy gradient objective. While the observed trends are consistent across our evaluated settings, broader evaluations on additional architectures, tasks, and diversity metrics would further strengthen the generality of our conclusions. Extending this study to larger-scale models, text-to-video generation, discrete diffusion language models, and broader generative modeling settings is an important direction for future work.

## Acknowledgments

The authors are grateful for partial supports from NSF Grants ECCS-1942523, DMS-2206576, 2409016, 2450378, and AFOSR Grant FA9550-25-1-0169. JC is supported by National Research Foundation of Korea (NRF) grants (RS-2024-00342044). YZ and MT gratefully acknowledge the partial supports by NSF Grant DMS-2513699 (YZ & MT), DOE Grants NA0004261 (MT), SC0026274 (YZ & MT), Richard Duke Fellowship (YZ & MT), and Simons Institute for the Theory of Computing at UC Berkeley (MT). WG acknowledges Georgia Tech ARC-ACO Fellowship for partial support.

## Impact Statement

This paper presents work aimed at advancing the field of generative models. It proposes a new approach to training improved image generators using reinforcement learning.

For example, it may be abused to create various harmful and offensive content. We strongly caution the community against such use cases.

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

# A. Proof of Thm. 3.1

**Derivation of Policy-Gradient Objectives** We show that EPG, PEPG, and PAR share the target optimal distribution as objective minimizers. We omit the condition $c$ as all probabilities are conditioned given a prompt $c$. We denote the policy of the pretrained model as $\pi_{\text{ref}}$. For convenience, we denote the advantage as

$$A(\boldsymbol{x}_0) := \frac{\eta}{\beta}\left(R(\boldsymbol{x}_0) - b\right).$$

The policy gradient losses (EPG), (PEPG), and (PAR) are the Monte Carlo simulation with $G$ samples of (3), (6) and (10), respectively, with the baseline given by $b = \mathbb{E}_{\boldsymbol{x}_0 \sim \pi_{\theta_{\text{old}}}}[R(\boldsymbol{x}_0)]$ and estimated by these samples.

**Exact Policy Gradient (EPG)** Given the $\pi_{\theta_{\text{old}}}$ and $\pi_{\text{ref}}$, we could reformulate (EPG) a loss functional of $\pi_\theta$ as follows:

$$\mathcal{L}_{\text{epg}}(\pi_\theta; \pi_{\theta_{\text{old}}}, \pi_{\text{ref}}) := \mathbb{E}_{\boldsymbol{x}_0 \sim \pi_{\theta_{\text{old}}}}\left[A(\boldsymbol{x}_0)\frac{\text{sg}\left(\pi_\theta\right)}{\pi_{\theta_{\text{old}}}}(\boldsymbol{x}_0)\log\pi_\theta(\boldsymbol{x}_0)\right] - \eta\,\text{KL}(\pi_\theta\,\|\,\pi_{\text{ref}}), \tag{3}$$

Note that the $\theta$-gradient of (3) is the same as the following loss:

$$\mathcal{L}_{\text{epg}}(\pi_\theta; \pi_{\theta_{\text{old}}}, \pi_{\text{ref}}) = \mathbb{E}_{\boldsymbol{x}_0 \sim \pi_{\theta_{\text{old}}}}\left[A(\boldsymbol{x}_0)\frac{\pi_\theta}{\pi_{\theta_{\text{old}}}}(\boldsymbol{x}_0)\right] - \eta\,\text{KL}(\pi_\theta\,\|\,\pi_{\text{ref}}). \tag{4}$$

By taking the first variation of $L[\pi_\theta] := \mathcal{L}_{\text{epg}}[\pi_\theta] + C(\int\pi_\theta(\boldsymbol{x}_0)\mathrm{d}\boldsymbol{x}_0 - 1)$ w.r.t. $\pi_\theta$ (where $C \in \mathbb{R}$ is the Lagrangian multiplier), we can obtain a explicit description of an optimal policy as follows:

$$\begin{aligned}0 = \frac{\delta L}{\delta\pi_\theta}(\pi_\theta; \boldsymbol{x}_0) &= A(\boldsymbol{x}_0) - \eta\log\frac{\pi_\theta}{\pi_{\text{ref}}}(\boldsymbol{x}_0) - \eta + C \\ &= \eta\left(\frac{1}{\beta}\left(R(\boldsymbol{x}_0) - b\right) - \log\frac{\pi_\theta}{\pi_{\text{ref}}}(\boldsymbol{x}_0) - 1 + \frac{C}{\eta}\right).\end{aligned} \tag{5}$$

By organizing (5), we obtain the optimal policy $\pi_{\text{epg}}^\star$ as follows:

$$\pi_{\text{epg}}^\star(\boldsymbol{x}_0 \mid \boldsymbol{c}) \propto \exp(R(\boldsymbol{x}_0 \mid \boldsymbol{c})/\beta)\pi_{\text{ref}}(\boldsymbol{x}_0 \mid \boldsymbol{c}).$$

**Proximal Exact Policy Gradient (PEPG)** Similarly, we mathematically reformulate (PEPG) as the following loss with the same $\theta$-gradient:

$$\mathcal{L}_{\text{pepg}}(\pi_\theta; \pi_{\theta_{\text{old}}}, \pi_{\text{ref}}) := \mathbb{E}_{\boldsymbol{x}_0 \sim \pi_{\theta_{\text{old}}}}\left[\left(A(\boldsymbol{x}_0) - \log\frac{\text{sg}\left(\pi_\theta\right)}{\pi_{\theta_{\text{old}}}}(\boldsymbol{x}_0)\right)\frac{\text{sg}\left(\pi_\theta\right)}{\pi_{\theta_{\text{old}}}}(\boldsymbol{x}_0)\log\pi_\theta(\boldsymbol{x}_0)\right] - \eta\,\text{KL}(\pi_\theta\,\|\,\pi_{\text{ref}}). \tag{6}$$

We adopt an iterative policy optimization scheme in which a sampling policy is maintained and updated across stages. At stage $k$, we denote the sampling policy by $\pi_k \equiv \pi_{\text{old}}$, which corresponds to the policy obtained from the previous iteration. Then, our optimization scheme can be written as follows:

$$\pi_{k+1} = \arg\min_{\pi_\theta}\mathcal{L}_{\text{pepg}}(\pi_\theta; \pi_k, \pi_{\text{ref}}), \quad \text{for } k \in \{1, 2, \dots\}. \tag{7}$$

This update admits a closed-form solution at the level of path measures:

$$\pi_{k+1}(\boldsymbol{x}_0 \mid \boldsymbol{c}) \propto \left(\exp(R(\boldsymbol{x}_0 \mid \boldsymbol{c})/\beta)\,\pi_{\text{ref}}(\boldsymbol{x}_0 \mid \boldsymbol{c})\right)^{\frac{\eta}{1+\eta}}\left(\pi_k(\boldsymbol{x}_0 \mid \boldsymbol{c})\right)^{\frac{1}{1+\eta}}. \tag{8}$$

*Proof.* Take the first variation of $L[\pi_\theta] := \mathcal{L}_{\text{pepg}}[\pi_\theta] + C(\int\pi_\theta(\boldsymbol{x}_0)\mathrm{d}\boldsymbol{x}_0 - 1)$ w.r.t. $\pi_\theta$ (where $C \in \mathbb{R}$ is the Lagrangian multiplier):

$$\begin{aligned}0 = \frac{\delta L}{\delta\pi_\theta}(\pi_\theta; \boldsymbol{x}_0) &= \left(A(\boldsymbol{x}_0) - \log\frac{\text{sg}\left(\pi_\theta\right)}{\pi_{\theta_{\text{old}}}}(\boldsymbol{x}_0)\right)\frac{\text{sg}\left(\pi_\theta\right)}{\pi_{\theta_{\text{old}}}}(\boldsymbol{x}_0)\frac{\pi_{\theta_{\text{old}}}}{\text{sg}\left(\pi_\theta\right)}(\boldsymbol{x}_0) - \eta\log\frac{\pi_\theta}{\pi_{\text{ref}}}(\boldsymbol{x}_0) - \eta + C \\ &= A(\boldsymbol{x}_0) - \log\frac{\pi_\theta}{\pi_{\theta_{\text{old}}}}(\boldsymbol{x}_0) - \eta\log\frac{\pi_\theta}{\pi_{\text{ref}}}(\boldsymbol{x}_0) - \eta + C \\ &= \frac{\eta}{\beta}\left(R(\boldsymbol{x}_0) - b\right) - \log\frac{\pi_\theta^{\eta+1}}{\pi_{\theta_{\text{old}}}\pi_{\text{ref}}^\eta}(\boldsymbol{x}_0) - \eta + C.\end{aligned} \tag{9}$$

By organizing (9), we obtain (8). $\qquad\square$

As $k \to \infty$, the sequence $\{\pi_k\}$ converges to the target path measure $\pi^\star(\boldsymbol{x}_0 \mid \boldsymbol{c}) \propto \exp(R(\boldsymbol{x}_0 \mid \boldsymbol{c})/\beta)\pi_{\text{ref}}(\boldsymbol{x}_0 \mid \boldsymbol{c})$. This result shows that the PEPG provides an exact and principled alternative to GRPO.

**Proximal Advantage Regression (PAR)**  Finally, we consider an advantage regression, or advantage matching, loss function in (PAR), which can be reformulated as follows:

$$\mathcal{L}_{\text{par}}(\pi_\theta; \pi_{\theta_{\text{old}}}, \pi_{\text{ref}}) := \mathbb{E}_{\boldsymbol{x}_0 \sim \pi_{\theta_{\text{old}}}} \left[ -\frac{1}{2} \frac{\text{sg}\,(\pi_\theta)}{\pi_{\theta_{\text{old}}}}(\boldsymbol{x}_0) \left\| A(\boldsymbol{x}_0) - \log \frac{\pi_\theta}{\pi_{\theta_{\text{old}}}}(\boldsymbol{x}_0) \right\|^2 \right] - \eta \, \text{KL}(\pi_\theta \parallel \pi_{\text{ref}}). \tag{10}$$

This loss penalizes deviations of the log-likelihood ratio from the advantage function via least-square regression. The key role of the stop-gradient operator $\text{sg}(\cdot)$ in (10) is to prevent the chain rule from producing an extra residual factor when computing the first variation. Specifically, treating $\text{sg}(\pi_\theta)/\pi_{\theta_{\text{old}}}$ as a constant weight, the first variation of the squared term with respect to $\pi_\theta$ yields only the inner derivative $-1/\pi_\theta$, which cancels with the $\text{sg}(\pi_\theta)/\pi_{\theta_{\text{old}}}$ prefactor (since $\text{sg}(\pi_\theta) = \pi_\theta$ at evaluation). This simplification makes the first-order optimality condition of PAR identical to that of PEPG, as shown in (12). We adopt an iterative policy optimization scheme in which a sampling policy is maintained and updated across stages. At stage $k$, we denote the sampling policy by $\pi_k \equiv \pi_{\text{old}}$, which corresponds to the policy obtained from the previous iteration. Then, our optimization scheme can be written as follows:

$$\pi_{k+1} = \underset{\pi_\theta}{\arg\min} \, \mathcal{L}_{\text{par}}(\pi_\theta; \pi_k, \pi_{\text{ref}}), \quad \text{for } k \in \{1, 2, \dots\}. \tag{11}$$

This update also admits the same closed-form solution (8).

*Proof.* Take the first variation for $L[\pi_\theta] := \mathcal{L}_{\text{par}}[\pi_\theta] + C(\int \pi_\theta(\boldsymbol{x}_0)\mathrm{d}\boldsymbol{x}_0 - 1)$ w.r.t. $\pi_\theta$ (where $C \in \mathbb{R}$ is the Lagrangian multiplier):

$$\begin{aligned}
0 = \frac{\delta L}{\delta \pi_\theta}(\pi_\theta; \boldsymbol{x}_0) &= \left( A(\boldsymbol{x}_0) - \log \frac{\text{sg}\,(\pi_\theta)}{\pi_{\theta_{\text{old}}}}(\boldsymbol{x}_0) \right) \frac{\text{sg}\,(\pi_\theta)}{\pi_{\theta_{\text{old}}}}(\boldsymbol{x}_0) \frac{\pi_{\theta_{\text{old}}}}{\text{sg}\,(\pi_\theta)}(\boldsymbol{x}_0) - \eta \log \frac{\pi_\theta}{\pi_{\text{ref}}}(\boldsymbol{x}_0) - \eta + C \\
&= A(\boldsymbol{x}_0) - \log \frac{\pi_\theta}{\pi_{\theta_{\text{old}}}}(\boldsymbol{x}_0) - \eta \log \frac{\pi_\theta}{\pi_{\text{ref}}}(\boldsymbol{x}_0) - \eta + C \\
&= \frac{\eta}{\beta} \left( R(\boldsymbol{x}_0) - b \right) - \log \frac{\pi_\theta^{\eta+1}}{\pi_{\theta_{\text{old}}} \pi_{\text{ref}}^\eta}(\boldsymbol{x}_0) - \eta + C.
\end{aligned} \tag{12}$$

By organizing (12), we obtain (8). □

As $k \to \infty$, the sequence $\{\pi_k\}$ converges to the target path measure $\pi^\star(\boldsymbol{x}_0 \mid \boldsymbol{c}) \propto \exp(R(\boldsymbol{x}_0 \mid \boldsymbol{c})/\beta)\pi_{\text{ref}}(\boldsymbol{x}_0 \mid \boldsymbol{c})$. This loss shares structural similarities with the objectives used in GFlowNet (Bengio et al., 2021) and related variants (Kimi Team et al., 2025; Malkin et al., 2022).

# B. Additional Technical Details

### B.1. ELBO weighting

Various ELBO objectives have been proposed for training diffusion and flow models effectively (Song et al., 2020; Kingma et al., 2021; Kingma & Gao, 2023; Karras et al., 2022; Shi & Titsias, 2025). The ELBO variants differ mostly in how they weight $w(t)$ and the regression objective (i.e., the $\boldsymbol{\epsilon}, \boldsymbol{x}, \boldsymbol{v}$-loss) (Li & He, 2025). For the simplicity of demonstration, we unified each objective considered in this work in the same form of $\boldsymbol{v}$-loss $\mathbb{E}\|\boldsymbol{v}_\theta - \boldsymbol{v}\|_2^2$ with different weighting. We considered several different ELBO.

**Path-KL weighting**: Following the original derivation in Score-based diffusion model (Song et al., 2020) that uses variational inference and simplifies the KL divergence between the forward and backward path measure, the ELBO is equivalent to with weighting $w(t) = \frac{1-t}{t}$ in $\boldsymbol{v}$-loss,

$$\text{ELBO}_{\text{path}}(\boldsymbol{v}_\theta, \boldsymbol{x}_0) = \mathbb{E}_{t, \boldsymbol{\epsilon}} \left[ \frac{1-t}{t} \left\| \boldsymbol{v}_\theta - \boldsymbol{v} \right\|_2^2 \right] \tag{13}$$

**Simple weighting**: Apart from path-KL weighting, constant weighting across all $t$ is also shown to achieve decent performance in diffusion training (Ho et al., 2020; Shi & Titsias, 2025). Following a similar intuition, we consider the following simply weighted ELBO with $w(t) = 1$,

$$\text{ELBO}_{\text{simple}}(\boldsymbol{v}_\theta, \boldsymbol{x}_0) = \mathbb{E}_{t,\boldsymbol{\epsilon}}\left[\left\|\boldsymbol{v}_\theta - \boldsymbol{v}\right\|_2^2\right] \tag{14}$$

**Adaptive weighting**: Besides time-dependent only weighting, prior works (Yin et al., 2024; Zheng et al., 2025c) have also adopted data-dependent weighting that self-normalizes the objective to ensure numerical robustness. We similarly consider such a formulation, express in $\boldsymbol{v}$-loss as,

$$\text{ELBO}_{\text{adapt}}(\boldsymbol{v}_\theta, \boldsymbol{x}_0) = \mathbb{E}_{t,\boldsymbol{\epsilon}}\left[\frac{t \cdot d \cdot \left\|\boldsymbol{v}_\theta - \boldsymbol{v}\right\|_2^2}{\text{sg}(\left\|\boldsymbol{v}_\theta - \boldsymbol{v}\right\|_1)}\right] \tag{15}$$

### B.2. Connection between methods

Existing works, such as AWM (Xue et al., 2025a) and FlowGRPO (Liu et al., 2025b), have also adapted GRPO to diffusion models and can be unified under a common lens of likelihood estimation. AWM uses ELBO (ELBO) with simple weighting (14) and a 1 Monte Carlo sample, computing probability ratio as,

$$\frac{\pi_\theta(\boldsymbol{x}_0)}{\pi_{\theta_{\text{old}}}(\boldsymbol{x}_0)} = \exp\left(\left\|\boldsymbol{v}_{\text{old}}(\boldsymbol{x}_t, t) - \boldsymbol{v}\right\|_2^2 - \left\|\boldsymbol{v}_\theta(\boldsymbol{x}_t, t) - \boldsymbol{v}\right\|_2^2\right)$$

FlowGRPO adopts the trajectory estimator (Trajectory), with the additional approximation that express probabilit ratio as using sum-of-exp rather than exp-of-sum,

$$\frac{\pi_\theta(\boldsymbol{x}_0)}{\pi_{\theta_{\text{old}}}(\boldsymbol{x}_0)} = \exp\left(\sum_{i=1}^N \log\frac{p_\theta(\boldsymbol{x}_{t_{i-1}}|\boldsymbol{x}_{t_i})}{p_{\text{old}}(\boldsymbol{x}_{t_{i-1}}|\boldsymbol{x}_{t_i})}\right)$$

$$\approx \sum_{i=1}^N \exp\left(\log\frac{p_\theta(\boldsymbol{x}_{t_{i-1}}|\boldsymbol{x}_{t_i})}{p_{\text{old}}(\boldsymbol{x}_{t_{i-1}}|\boldsymbol{x}_{t_i})}\right) = \sum_{i=1}^N \frac{p_\theta(\boldsymbol{x}_{t_{i-1}}|\boldsymbol{x}_{t_i})}{p_{\text{old}}(\boldsymbol{x}_{t_{i-1}}|\boldsymbol{x}_{t_i})}$$

Moreover, Flow-GRPO applies a clipping operation to each term in the sum individually. These differences distinguish AWM and FlowGRPO from our adapted version of GRPO reported in Tab. 1.

### B.3. Details on ELBO and Loss computation

**ELBO Computation**  We consider several practical strategies for computing the ELBO-based likelihood estimators introduced in (13), (14), and (15). In principle, an accurate ELBO estimate requires approximating expectations over diffusion time and noise. Specifically, for a given clean sample $\boldsymbol{x}_0$, the ELBO involves the expectation

$$\mathbb{E}_{t,\boldsymbol{\epsilon}}\left[\left\|\boldsymbol{v}_\theta(\boldsymbol{x}_t, t) - \boldsymbol{v}(\boldsymbol{x}_t, t)\right\|^2\right],$$

where $(t, \boldsymbol{\epsilon}) \sim U[0,1] \times \mathcal{N}(\boldsymbol{0}, \boldsymbol{I})$ and $\boldsymbol{x}_t = (1-t)\boldsymbol{x}_0 + t\boldsymbol{\epsilon}$. A straightforward Monte Carlo approximation draws $M$ independent samples $\{(t_i, \boldsymbol{\epsilon}_i)\}_{i=1}^M$ and estimates the expectation as

$$\mathbb{E}_{t,\boldsymbol{\epsilon}}\left[\left\|\boldsymbol{v}_\theta(\boldsymbol{x}_t, t) - \boldsymbol{v}(\boldsymbol{x}_t, t)\right\|^2\right] \approx \frac{1}{M}\sum_{i=1}^M \left\|\boldsymbol{v}_\theta(\boldsymbol{x}_{t_i}, t_i) - \boldsymbol{v}(\boldsymbol{x}_{t_i}, t_i)\right\|^2, \tag{16}$$

with $\boldsymbol{x}_{t_i} = (1-t_i)\boldsymbol{x}_0 + t_i\boldsymbol{\epsilon}_i$. While this estimator is unbiased, evaluating the ELBO using multiple samples per data point can be computationally expensive. To reduce this cost, we consider two simplified Monte Carlo schemes that trade variance for efficiency:

- **Single-timestep estimation.** Let the diffusion time interval be discretized as $T := \{t_1 := 1/N, \ldots, t_N := 1\}$ (see Alg. 1). For a given timestep $t_i \in T$ and noise sample $\boldsymbol{\epsilon} \sim \mathcal{N}(\boldsymbol{0}, \boldsymbol{I})$, we define $\boldsymbol{x}_{t_i} = (1-t_i)\boldsymbol{x}_0 + t_i\boldsymbol{\epsilon}$. The ELBO expectation is then approximated using a single Monte Carlo sample:

$$\mathbb{E}_{t,\boldsymbol{\epsilon}}\left[\left\|\boldsymbol{v}_\theta(\boldsymbol{x}_t, t) - \boldsymbol{v}(\boldsymbol{x}_t, t)\right\|^2\right] \approx \left\|\boldsymbol{v}_\theta(\boldsymbol{x}_{t_i}, t_i) - \boldsymbol{v}(\boldsymbol{x}_{t_i}, t_i)\right\|^2. \tag{17}$$

For instance, under the simple weighting scheme in (14), the importance-weighted policy-gradient term can be approximated as

$$\frac{\text{sg}(\pi_\theta)}{\pi_{\theta_{\text{old}}}}(\boldsymbol{x}_0)\log\pi_\theta(\boldsymbol{x}_0) \approx -\frac{\exp\Big(-\|\boldsymbol{v}_\theta(\boldsymbol{x}_{t_i},t_i)-\boldsymbol{v}(\boldsymbol{x}_{t_i},t_i)\|^2\Big)}{\exp\Big(-\|\boldsymbol{v}_{\text{old}}(\boldsymbol{x}_{t_i},t_i)-\boldsymbol{v}(\boldsymbol{x}_{t_i},t_i)\|^2\Big)}\,\|\boldsymbol{v}_\theta(\boldsymbol{x}_{t_i},t_i)-\boldsymbol{v}(\boldsymbol{x}_{t_i},t_i)\|^2. \tag{18}$$

- **All-timestep estimation.** An alternative approach estimates the ELBO by aggregating contributions across all discretized timesteps:

$$\mathbb{E}_{t,\boldsymbol{\epsilon}}\Big[\|\boldsymbol{v}_\theta(\boldsymbol{x}_t,t)-\boldsymbol{v}(\boldsymbol{x}_t,t)\|^2\Big] \approx \frac{1}{N}\sum_{i=1}^{N}\|\boldsymbol{v}_\theta(\boldsymbol{x}_{t_i},t_i)-\boldsymbol{v}(\boldsymbol{x}_{t_i},t_i)\|^2. \tag{19}$$

In this case, the importance ratio $\frac{\text{sg}(\pi_\theta)}{\pi_{\theta_{\text{old}}}}(\boldsymbol{x}_0)$ can be precomputed by evaluating the ELBO across the full set of timesteps using shared noise realizations. The resulting ratio is then reused across all timestep contributions along the same trajectory.

Although the all-timestep estimator provides a more faithful Monte Carlo approximation of the ELBO, our empirical results in Fig. 5 show that both estimators achieve comparable performance. Given its lower computational cost and simpler implementation, we therefore recommend the single-timestep estimator as a practical default.

**KL Divergence** Now, we describe how we compute KL divergence between $\pi_\theta$ and the reference policy $\pi_{\text{ref}}$ in a closed form. Here, we assume that $\pi_{\theta_{\text{old}}} \approx \pi_\theta$, and solve

$$\text{KL}(\pi_\theta \,\|\, \pi_{\text{ref}}) = \mathbb{E}_t\mathbb{E}_{\boldsymbol{x}_t}\left[\frac{1}{2}\left(\frac{g_t(1-t)}{2t}+\frac{1}{g_t}\right)^2\|\boldsymbol{v}_\theta(\boldsymbol{x}_t,t)-\boldsymbol{v}_{\text{ref}}(x_t,t)\|^2\right] = \mathbb{E}_t\mathbb{E}_{\boldsymbol{x}_t}w(t)\|\boldsymbol{v}_\theta(\boldsymbol{x}_t,t)-\boldsymbol{v}_{\text{ref}}(x_t,t)\|^2, \tag{20}$$

where $w(t) = a^2\frac{1-t}{t}$. We simply use simple approximation, i.e. $w(t) = 1$ in practice, following (Zheng et al., 2025c; Xue et al., 2025a).

## C. Implementation Details

**Algorithm** Alg. 1 presents a unified framework for reward-based diffusion fine-tuning that accommodates different policy-gradient objectives, likelihood estimators, and sampling strategies. Starting from a pretrained diffusion velocity field $v$, we iteratively improves the model by **(1) sampling** data $\boldsymbol{x}_0$ from a reference (old) policy $\pi_{\theta_{\text{old}}}$ (i.e. sample with $\boldsymbol{v}_{\text{old}}$), **(2) computing likelihood** by trajectory-based or ELBO-based estimator, and **(3)** updating the current policy using a chosen **policy-gradient loss**. After updating the model parameters, the old policy is updated via an exponential moving average (EMA) of the current parameters, ensuring a slowly evolving reference policy that stabilizes optimization. This unified procedure enables efficient and stable fine-tuning across a broad class of objectives and likelihood estimation schemes. The ema decay rate is $\alpha_i$ where $i$ is the number of current epoch. Moreover, note that sampling batch over training mini-batch is the number of gradient steps per epoch.

**Hyperparameter Settings** Our experimental setup largely follows DiffusionNFT (Zheng et al., 2025c) and FlowGRPO (Liu et al., 2025b). For each epoch, we use 48 prompts, and 24 rollouts (or group size) per each prompts. We use LoRA (Hu et al., 2022) configuration of $\alpha = 64$, $r = 32$, and learning rate ($3 \times 10^{-4}$). For each collected clean image, forward noising and loss computation are performed exactly at the corresponding sampling timesteps. We employ a second-order ODE sampler for data collection and enable adaptive time weighting by default. We follow the same KL divergence estimation procedure as FlowGRPO and DiffusionNFT, approximated through the Girsanov theorem. For each epoch, we use 48 prompts. For experiments involving SDE-based sampling, we use a noise level of 0.7. We further adopt the same exponential moving average (EMA) decay scheme as DiffusionNFT, where *decay type 1* is $\alpha_i = \min(0.001i, 0.5)$, and *decay type 2* is $\alpha_i = \min(0.01i, 0.8)$ in Alg. 1. Moreover, among all configurations, only the EPG objective combined with ELBO-based likelihood estimation and SDE sampling required gradient clipping (with a threshold of 0.01) to ensure training stability. Gradient clipping was not applied in other settings. Additional hyperparameter configurations are summarized in Tab. 3.

---

**Algorithm 1** Unified reward-based diffusion fine-tuning algorithm

---

**Require:** Loss $\mathcal{L} \in \{(\text{EPG}), (\text{PEPG}), (\text{PAR})\}$, likelihood estimator $\text{LIKELIHOOD}(\boldsymbol{v}, \boldsymbol{x}, t, \boldsymbol{c})$, data sampler $\boldsymbol{x}_0 = \text{SAMPLER}(\boldsymbol{v}, \boldsymbol{x}_1, \boldsymbol{c})$.

**Require:** Pretrained velocity $\boldsymbol{v}_{\text{ref}}$, reward $r(\cdot)$, scale $\beta$, regularization $\eta$, decay type $\{\alpha_i\}$, timesteps $T = \{1/N, 2/N, \ldots, 1\}$.

1: Initialize $\boldsymbol{v}_\theta \leftarrow \boldsymbol{v}_{\text{ref}}$.
2: **for** $i \in \{1, 2, \ldots\}$ **do**
3:      Sample a batch of prompts $\boldsymbol{c}$ and $\boldsymbol{x}_1 \sim \mathcal{N}(\boldsymbol{0}, \boldsymbol{I})$.
4:      Sample a batch of data $\boldsymbol{x}_0 \leftarrow \text{SAMPLER}(\boldsymbol{v}_{\text{old}}, \boldsymbol{x}_1, \boldsymbol{c})$.
5:      Sample data by $\boldsymbol{x}_0 \leftarrow \text{SAMPLER}(\boldsymbol{v}_{\text{old}}, \boldsymbol{x}_1, \boldsymbol{c})$.
6:      **for** minibatch **do**
7:          **for** $t \in T$ **do**
8:              Compute $\pi_{\theta_{\text{old}}}(\boldsymbol{x}_0 \mid \boldsymbol{c}) \leftarrow \text{LIKELIHOOD}(\boldsymbol{v}_{\text{old}}, \boldsymbol{x}_0, t, \boldsymbol{c})$.
9:              Compute $\pi_\theta(\boldsymbol{x}_0 \mid \boldsymbol{c}) \leftarrow \text{LIKELIHOOD}(\boldsymbol{v}_\theta, \boldsymbol{x}_0, t, \boldsymbol{c})$.
10:             Accumulate the loss $\mathcal{L}$ with estimated $\pi_\theta(\boldsymbol{x}_0 \mid \boldsymbol{c})$ and $\pi_{\theta_{\text{old}}}(\boldsymbol{x}_0 \mid \boldsymbol{c})$.
11:          **end for**
12:          Update $\boldsymbol{v}_\theta$.
13:      **end for**
14:      Update $\theta_{\text{old}} \leftarrow (1 - \alpha_i)\theta + \alpha_i\theta_{\text{old}}$.
15: **end for**

---

**Reward Functions** For experiments on the GenEval (Ghosh et al., 2023) benchmark, we use the GenEval score as the sole reward signal. For the OCR task, we combine an OCR-based reward with human preference rewards, including PickScore (Kirstain et al., 2023), CLIPScore (Hessel et al., 2021), and HPSv2.1 (Wu et al., 2023). For experiments on the OCR benchmark, we further consider a composite reward constructed by aggregating PickScore, CLIPScore, and HPSv2.1. To account for differences in scale across reward functions, we rescale PickScore by a factor of $1/26$, following standard practice, so that its magnitude is comparable to the other reward terms. After normalization, all rewards (OCR, PickScore, CLIPScore, and HPSv2.1) are combined with equal weights. We also conduct experiments on the PickScore dataset using multiple reward functions, including PickScore, CLIPScore, and HPSv2.1, and apply the same weighting scheme.

**Other Benchmarks** We implement FlowGRPO, DiffusionNFT, and AWM based on their official GitHub repositories. For DiffusionNFT, we adapt the reward functional to our single-stage training setting by using a weighted sum of multiple rewards, whereas the original implementation employs a multi-stage training procedure. For AWM, we use the reported hyperparameters and disable classifier-free guidance (CFG) to ensure a fair comparison with our CFG-free training setup. Unless otherwise specified, we follow the default configurations provided by each method. Key hyperparameter settings are summarized in Tab. 4 for clarity.

**Samplers** Among various ODE-based samplers (Lu et al., 2022; Zhang & Chen, 2022), we adopt DPM (Lu et al., 2022). For SDE-based sampling, we use the standard Euler–Maruyama scheme. Unless otherwise specified, we use 10 steps for ODE samplers and 40 steps for SDE samplers in our implementations.

**Evaluation Metrics** We evaluate all models using a suite of standard reward and quality metrics, including GenEval (Ghosh et al., 2023), PickScore (Kirstain et al., 2023), CLIPScore (Hessel et al., 2021), HPSv2.1 (Wu et al., 2023), Aesthetic Score (Schuhmann, 2022), and ImageReward (Xu et al., 2023), following established evaluation protocols. Our baseline comparisons are SD3.5-M (Esser et al., 2024), SD-XL, SD3.5-L, DALLE-3 (Betker et al., 2023), GPT-4o (Achiam et al., 2023), and FLUX.1 Dev (Black Forest Labs, 2024). We evaluate all methods on GenEval and OCR using classifier-free guidance (CFG) (Ho & Salimans, 2022) with a scale of 4.5, and on DrawBench without CFG.

## D. Additional Experimental Results

As shown in Tab. 5, our method achieves the highest overall GenEval score of 0.96, matching or exceeding all baselines across individual sub-tasks. Notably, our approach attains perfect or near-perfect scores on Single Object (1.00), Two Objects (0.99), and Position (0.99), while achieving the best Color score (0.97) by a notable margin. The Counting (0.98)

*Table 3.* **Hyperparameter Settings** for training our methods.

| position | task | loss | sampler | # steps | $\eta$ | $\beta$ | $w(t)$ | decay type | # epochs | # grad./epoch |
|---|---|---|---|---|---|---|---|---|---|---|
| Tab. 1 (r.4) | GenEval | (EPG) | SDE | 40 | $10^{-4}$ | $10^{-3}$ | adaptive | 1 | 360 | 1 |
| Tab. 1 (r.5) | GenEval | (EPG) | ODE | 10 | $10^{-4}$ | $10^{-3}$ | adaptive | 1 | 360 | 2 |
| Tab. 1 (r.6) | GenEval | (PEPG) | SDE | 40 | $10^{-4}$ | $10^{-3}$ | adaptive | 1 | 360 | 2 |
| Tab. 1 (r.7) Tab. 5 (r.10) Tab. 2 (r.9) Fig. 5 | GenEval | (PEPG) | ODE | 10 | $10^{-4}$ | $10^{-3}$ | adaptive | 1 | 360 | 2 |
| Tab. 1 (r.8) | GenEval | (PAR) | SDE | 40 | $10^{-4}$ | $10^{-3}$ | adaptive | 1 | 360 | 1 |
| Tab. 1 (r.9) | GenEval | (PAR) | ODE | 10 | $10^{-4}$ | $10^{-3}$ | adaptive | 1 | 360 | 2 |
| Tab. 2 (r.13) | OCR | (PEPG) | ODE | 10 | $10^{-4}$ | $10^{-3}$ | adaptive | 2 | 70 | 2 |
| Tab. 2 (r.16) | Multi-reward | (PEPG) | ODE | 25 | $10^{-4}$ | $10^{-4}$ | adaptive | 2 | 500 | 2 |
| Fig. 5 | GenEval | (PEPG) | ODE | 10 | $10^{-4}$ | $10^{-3}$ | simple path-KL | 1 | 360 | 2 |
| Fig. 5 | GenEval | (PEPG) | ODE | 10 | $10^{-4}$ | $10^{-3}$ | simple, $t \in [0,1]$ path-KL, $t \in [0,1]$ adaptive, $t \in [0,1]$ | 1 | 360 | 1 |

*Table 4.* **Hyperparameter Settings** for other benchmarks.

| Task | method | cfg | sampler | NFEs | noise sched. | # epochs | # prompts/epoch | # grad./epoch |
|---|---|---|---|---|---|---|---|---|
| GenEval | Ours (ODE) | - | ODE | 10 | - | 360 | 48 | 2 |
| | Ours (SDE) | - | SDE | 40 | 0.7 | 360 | 48 | 1 |
| | DiffusionNFT | - | ODE | 10 | - | 500 | 48 | 1 |
| | FlowGRPO | 4.5 | SDE | 10 | 0.7 | 4100 | 48 | 2 |
| | AWM | - | ODE | 14 | - | 200 | 72 | 1 |
| OCR | Ours (ODE) | - | ODE | 10 | - | 70 | 48 | 2 |
| | DiffusionNFT | - | ODE | 10 | - | 70 | 48 | 1 |
| | AWM | - | ODE | 14 | - | 100 | 72 | 1 |

and Attribute (0.87) sub-tasks also match the strongest baseline (DiffusionNFT), indicating that our ELBO-based method provides uniformly strong compositional generation without specializing to any particular sub-task. Among baselines, AWM shows relatively weaker performance on Position (0.82) and Attribute (0.59), suggesting that its optimization strategy may be less effective for spatially structured prompts.

Regarding the efficiency comparison between ELBO+ODE and trajectory+SDE, we report the GPU hours to reach GenEval $\geq 0.95$ across 3 independent runs in Tab. 8. ELBO+ODE reaches this target in approximately 90–100 GPU hours, compared to $750.3 \pm 69.2$ for EPG (Traj.) and 423 for FlowGRPO, both of which use the trajectory estimator with SDE sampling. This gap is far too large to be attributed to seed variance and represents a qualitatively different regime of computational cost.

*Table 5.* **Performance comparison on GenEval.** We compare our method equipped with the proximal policy-gradient objective in (PEPG), ELBO-based likelihood estimation, and ODE sampling against prior RL-based diffusion fine-tuning methods and baseline models. As shown in the table, our approach achieves consistently strong performance across GenEval sub-tasks, matching or exceeding existing methods under a unified training configuration.

| Model | Single Obj. | Two Obj. | Counting | Color | Position | Attr | Overall |
|---|---|---|---|---|---|---|---|
| DALL·E-3 | 0.96 | 0.87 | 0.47 | 0.83 | 0.43 | 0.45 | 0.67 |
| GPT-4o | 0.99 | 0.92 | 0.85 | 0.92 | 0.75 | 0.61 | 0.84 |
| SD-XL | 0.98 | 0.74 | 0.39 | 0.85 | 0.15 | 0.23 | 0.55 |
| FLUX.1-Dev | 0.98 | 0.81 | 0.74 | 0.79 | 0.22 | 0.45 | 0.66 |
| SD3.5-L | 0.98 | 0.89 | 0.73 | 0.83 | 0.34 | 0.47 | 0.71 |
| SD3.5-M | 0.98 | 0.78 | 0.50 | 0.81 | 0.24 | 0.52 | 0.63 |
| Flow-GRPO | **1.00** | **0.99** | 0.95 | 0.92 | **0.99** | 0.86 | 0.95 |
| AWM | 0.98 | 0.90 | 0.90 | 0.91 | 0.82 | 0.59 | 0.89 |
| DiffusionNFT | **1.00** | **0.99** | **0.98** | 0.93 | 0.96 | **0.87** | 0.95 |
| Ours | **1.00** | **0.99** | **0.98** | **0.97** | **0.99** | **0.87** | **0.96** |

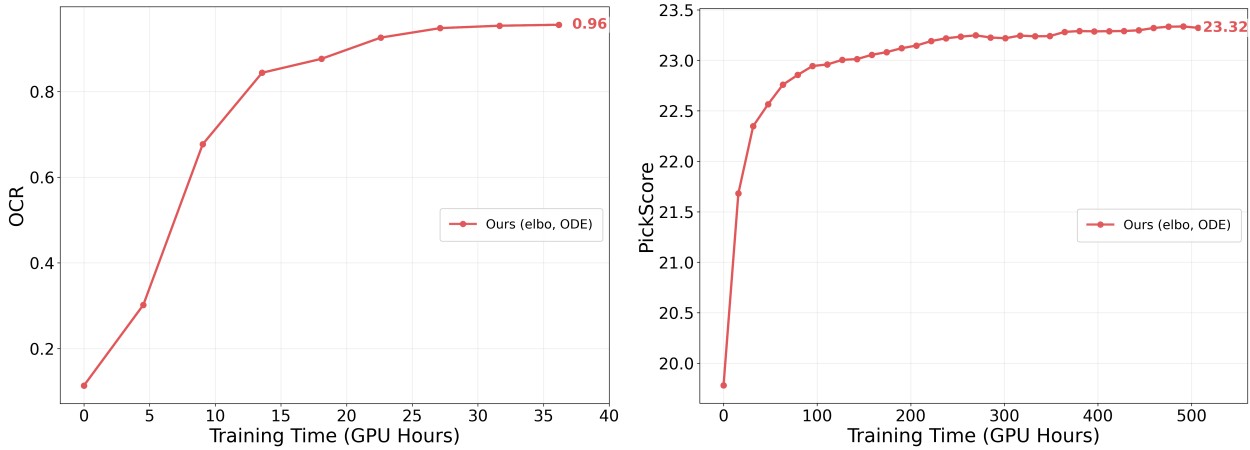

*Figure 6.* Performance of `OCR` (*left*) and `PickScore` (*right*) across training time.

*Table 6.* **Generalization evaluation on unseen GenEval categories.** We evaluate our method on unseen object classes and unseen counting scenarios from GenEval to assess generalization beyond the training distribution. DrawBench results (Tab. 2) additionally serve as a generalization test, as it contains prompts outside the GenEval training distribution.

| Method | Overall | Single Obj. | Two Obj. | Counting | Color | Position |
|---|---|---|---|---|---|---|
| SD3.5-M | 0.64 | 0.96 | 0.73 | 0.53 | 0.87 | 0.26 |
| +Flow-GRPO | 0.90 | 1.00 | 0.94 | 0.86 | 0.97 | 0.84 |
| +Ours (ELBO, ODE) | **0.93** | **1.00** | **0.96** | **0.91** | **0.99** | **0.91** |

*Table 7.* **Evaluation on FLUX.1-Dev.** To demonstrate that our approach generalizes beyond SD3.5-Medium, we apply our method (PEPG with ELBO-based likelihood estimation and ODE sampling) to FLUX.1-Dev and report results across standard evaluation metrics.

| Model | Method | GenEval | PickScore | ClipScore | HPSv2.1 | Aesthetic | ImgRwd |
|---|---|---|---|---|---|---|---|
| FLUX.1-Dev (w/ CFG) | Baseline | 0.66 | 22.84 | 0.295 | 0.274 | 5.71 | 0.96 |
| FLUX.1-Dev | Ours (ELBO, ODE) | **0.93** | **23.29** | **0.304** | **0.279** | 5.60 | **1.18** |

| SD3.5-M (w/o CFG) | SD3.5-M (w/ CFG) | FlowGRPO | Ours (elbo, ODE) |
|---|---|---|---|
| 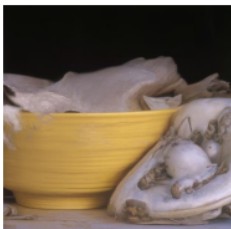 | 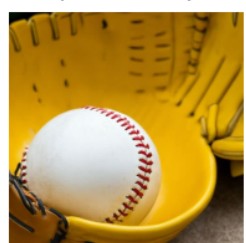 | 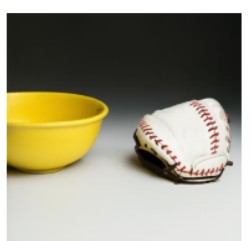 | 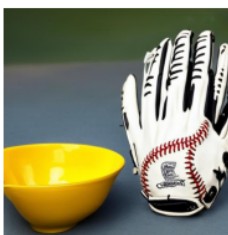 |

a photo of a yellow bowl and a white baseball glove

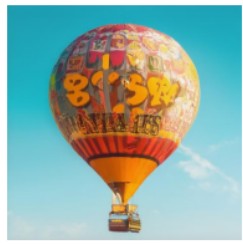 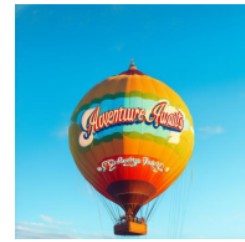 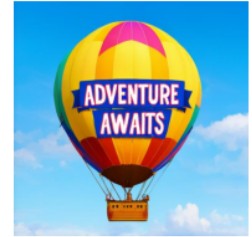 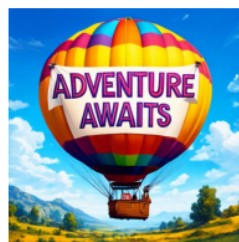

A vibrant hot air balloon ascends into a clear blue sky, trailing a banner that reads "Adventure Awaits" in bold, flowing letters. The balloon's

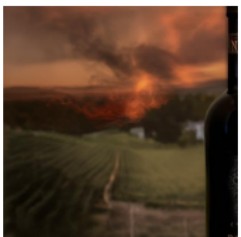 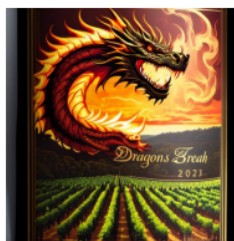 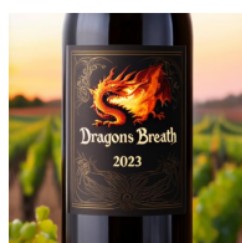 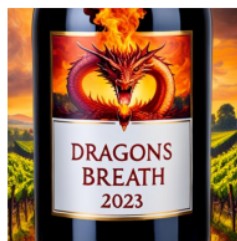

A close-up of a wine bottle with an intricate label titled "Dragons Breath 2023", featuring a fiery dragon exhaling smoke over a vineyard

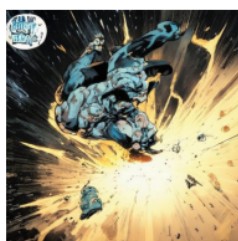 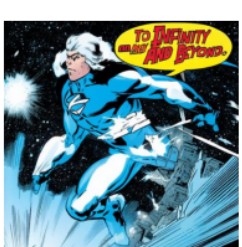 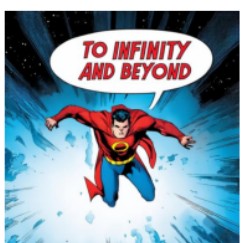 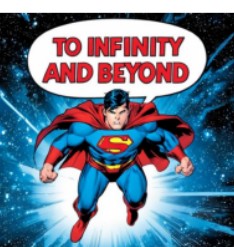

A vibrant comic book panel featuring a dynamic superhero leaping into the sky, with a bold speech bubble reading "To Infinity And Beyond"

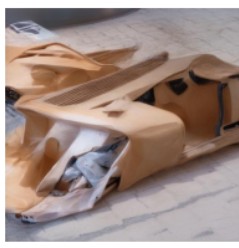 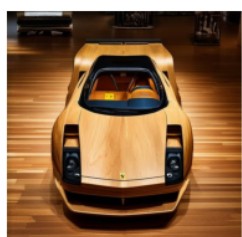 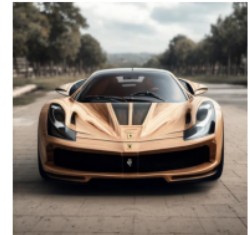 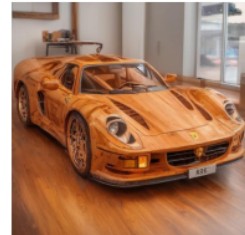

a Ferrari car that is made out of wood

*Figure 7.* Qualitative comparison between benchmarks and our model on Geneval, OCR, and PickScore prompts.

| SD3.5-M (w/o CFG) | SD3.5-M (w/ CFG) | FlowGRPO | Ours (elbo, ODE) |
|---|---|---|---|
| 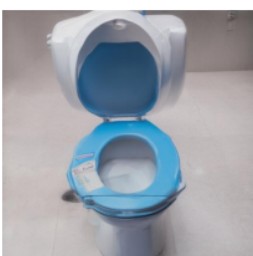 | 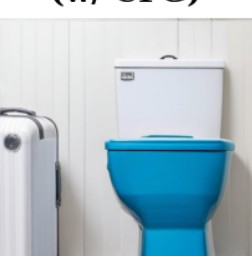 | 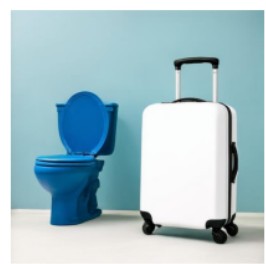 | 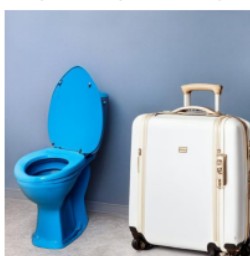 |

a photo of a blue toilet and a white suitcase

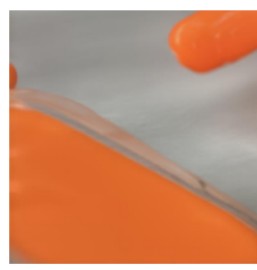 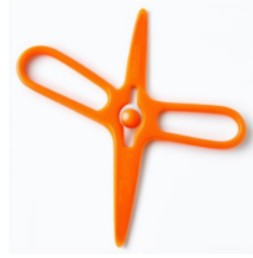 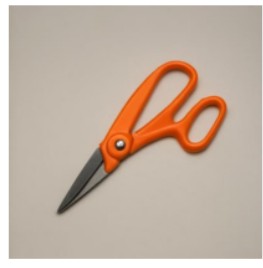 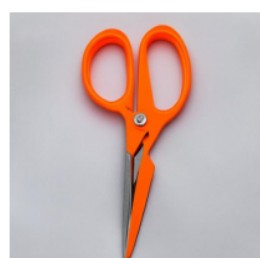

a photo of an orange scissors

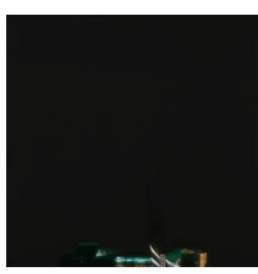 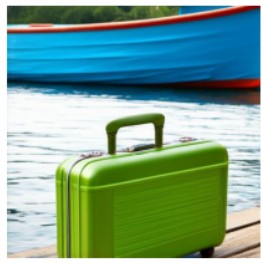 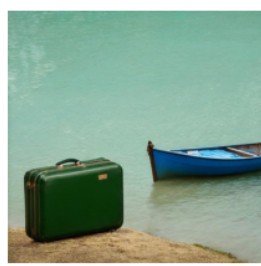 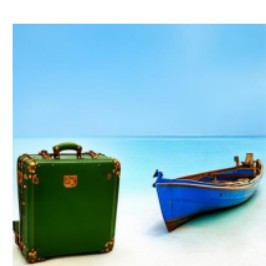

a photo of a green suitcase and a blue boat

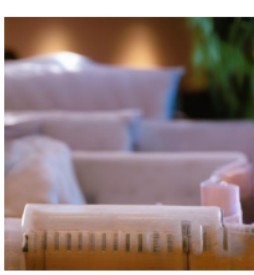 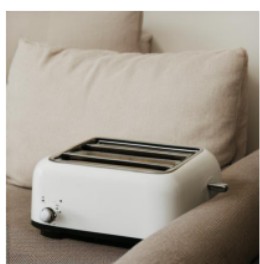 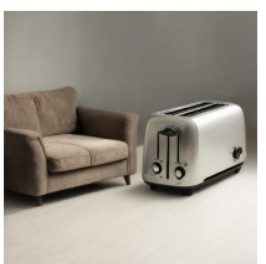 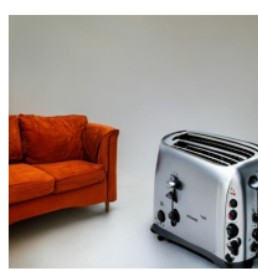

a photo of a couch left of a toaster

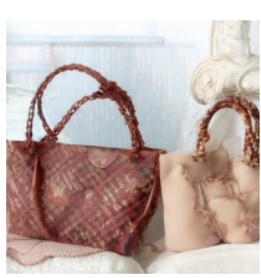 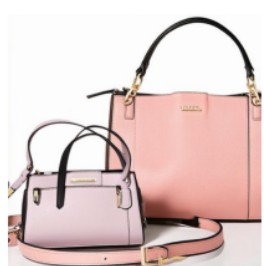 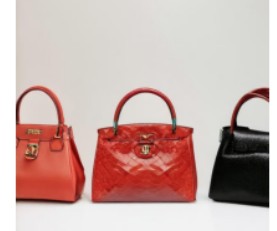 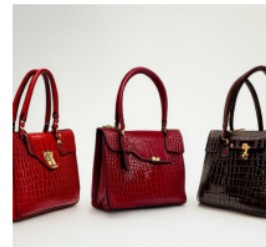

a photo of three handbags

*Figure 8.* Qualitative comparison between benchmarks and our model on GenEval prompts.

| SD3.5-M (w/o CFG) | SD3.5-M (w/ CFG) | FlowGRPO | Ours (elbo, ODE) |
|---|---|---|---|
| 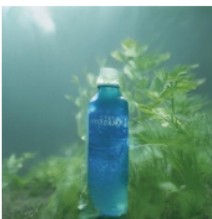 | 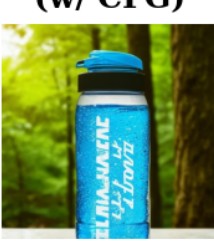 | 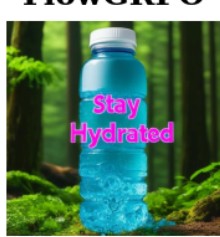 | 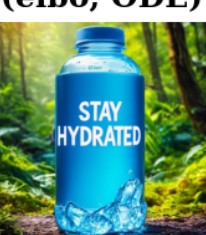 |

A vibrant, modern advertisement featuring a sleek water bottle with the slogan "Stay Hydrated" prominently displayed. The bottle is half-filled

| 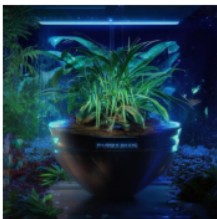 | 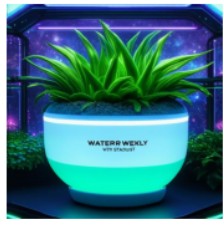 | 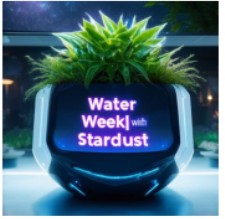 | 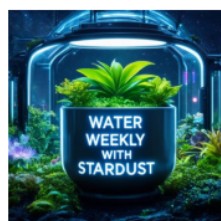 |
|---|---|---|---|

A futuristic greenhouse featuring an alien plant pot labeled "Water Weekly with Stardust", surrounded by bioluminescent flora and

| 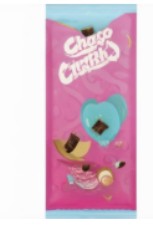 | 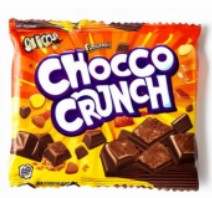 | 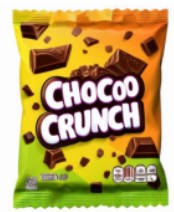 | 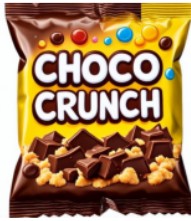 |
|---|---|---|---|

A vibrant candy wrapper design featuring "Choco Crunch", with a playful, colorful background and the brand name prominently displayed

| 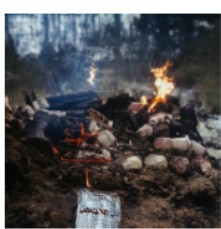 | 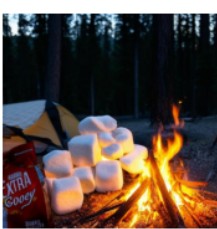 | 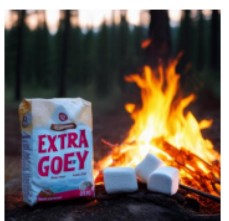 | 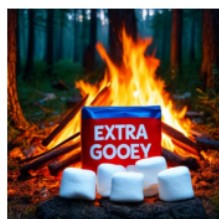 |
|---|---|---|---|

A cozy campsite at dusk, with a campfire blazing warmly. A package of marshmallows labeled "Extra Gooey" sits next to the fire, partially

| 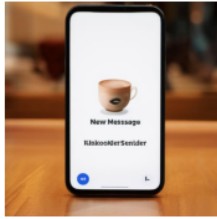 | 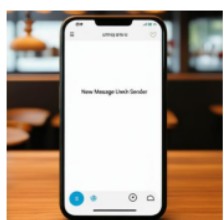 | 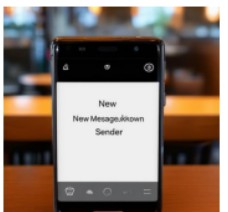 | 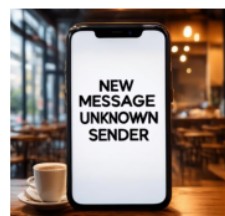 |
|---|---|---|---|

A realistic smartphone screen with a notification popping up, displaying "New Message Unknown Sender" in a modern, sleek interface, set

*Figure 9.* Qualitative comparison between benchmarks and our model on OCR prompts.

| SD3.5-M (w/o CFG) | SD3.5-M (w/ CFG) | FlowGRPO | Ours (elbo, ODE) |
|---|---|---|---|

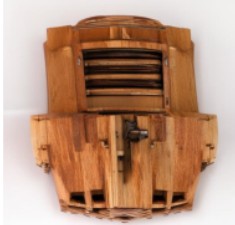 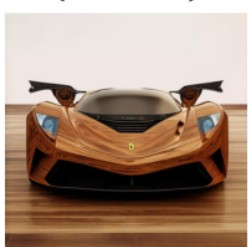 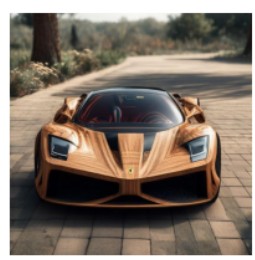 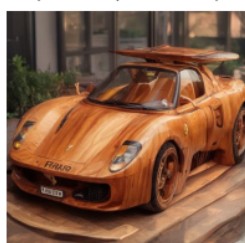

a Ferari car that is made out of wood

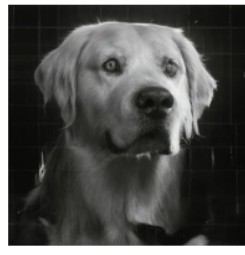 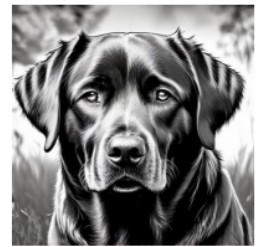 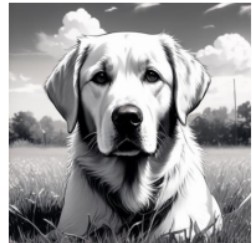 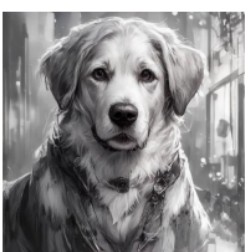

Labrador in anime style, black and white photography, highly detailed,

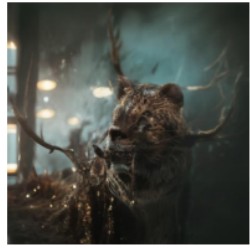 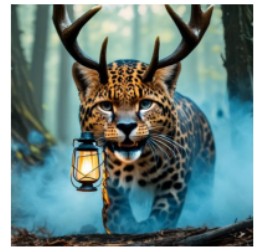 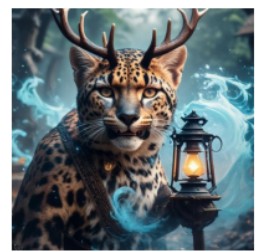 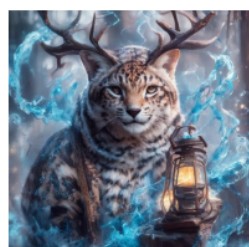

professional hyperrealistic 4k fantasy video game screenshot of a
hybrid between a bobcat ocelot and clouded leopard with antlers,

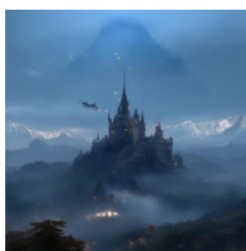 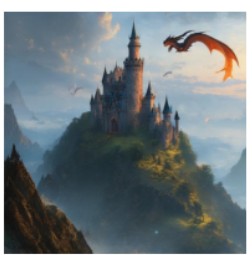 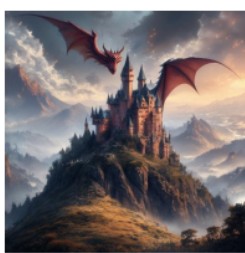 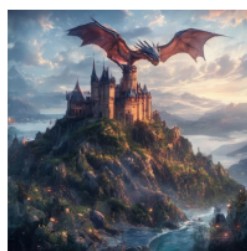

A fantasy landscape with a castle on a hill and a dragon flying overhead

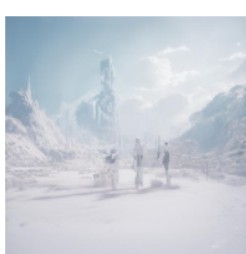 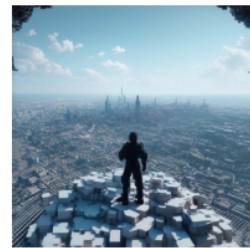 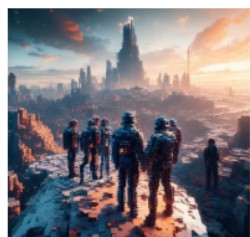 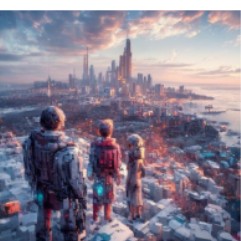

wideangle photo view futuristic people standing on voxel landscape
looking at a distant view ,with a city in the distance

*Figure 10.* Qualitative comparison between benchmarks and our model on PickScore prompts.

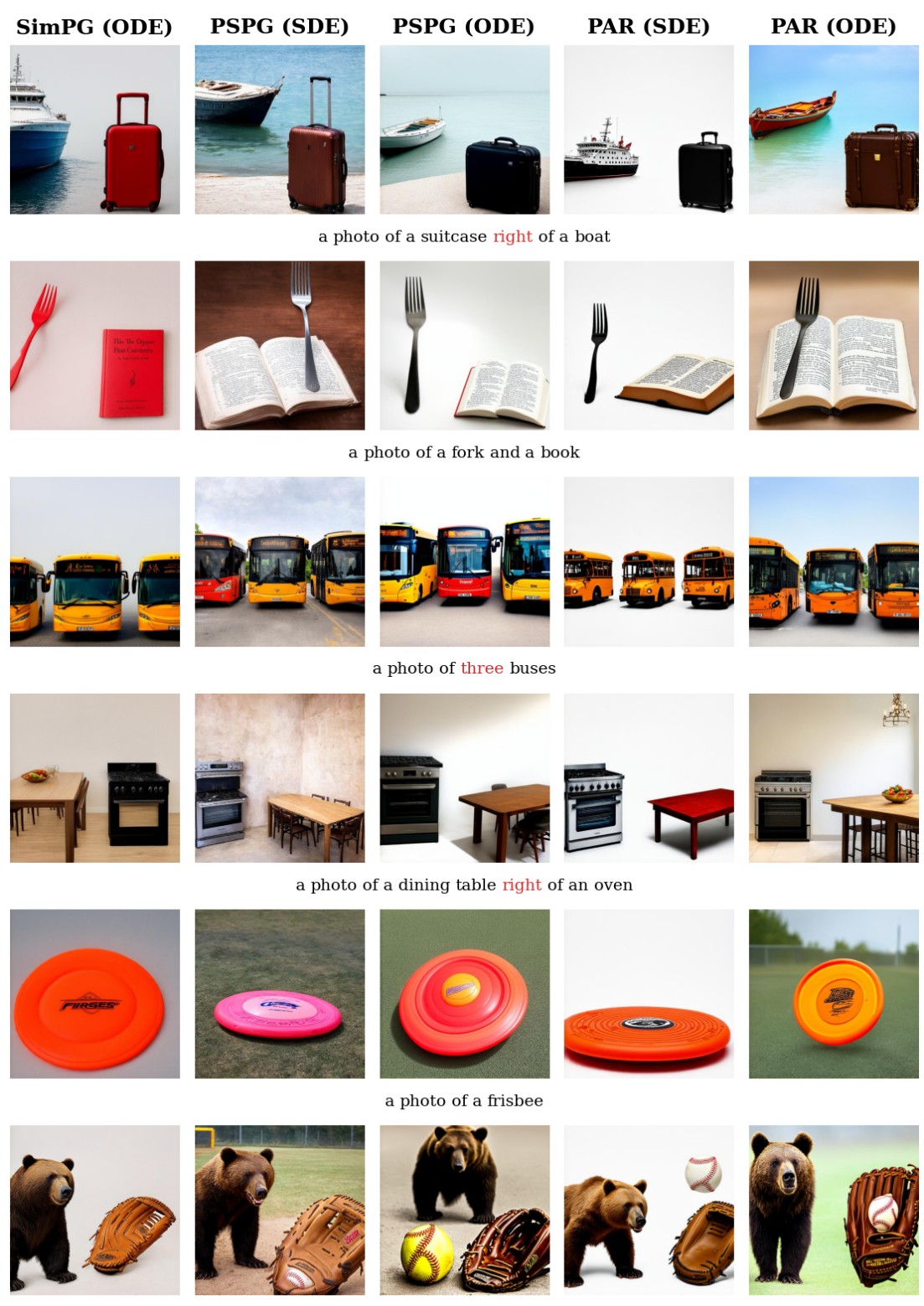

*Figure 11.* Qualitative comparison between ELBO-based Likelihood Estimation across variety of loss and samplers.

*Table 8.* **Multi-seed results (mean $\pm$ std over 3 runs).** Our original single-seed reporting followed the evaluation protocol of FlowGRPO, DiffusionNFT, and AWM, which also report single-seed results. We conducted 3 independent runs for PEPG and PAR in the ELBO+ODE setting. Both methods show very low variance across seeds, and the confidence intervals fully overlap, which directly supports our main claim: different loss formulations achieve comparable performance, and the precise choice of loss is not the dominant factor.

| Loss | GenEval | PickScore | ClipScore | HPSv2.1 | Aesthetic |
|------|---------|-----------|-----------|---------|-----------|
| PEPG | $0.95 \pm 0.01$ | $22.55 \pm 0.35$ | $0.302 \pm 0.004$ | $0.269 \pm 0.018$ | $5.35 \pm 0.06$ |
| PAR | $0.96 \pm 0.00$ | $22.71 \pm 0.24$ | $0.302 \pm 0.001$ | $0.263 \pm 0.009$ | $5.35 \pm 0.06$ |

*Table 9.* **GPU hours to reach GenEval $\geq$ 0.95.** Computational cost to reach GenEval $\geq$ 0.95. ELBO-based methods with an ODE sampler reach the target score using substantially fewer GPU hours than trajectory-based EPG.

|  | PEPG (ELBO, ODE) | PAR (ELBO, ODE) | EPG (Traj.) |
|--|-----------------|-----------------|-------------|
| GPU Hours | $90.0 \pm 4.5$ | $101.7 \pm 9.6$ | $750.3 \pm 69.2$ |
| GenEval | 0.95 | 0.95 | 0.93 |

