# OpenReview forum: "Rethinking the Design Space of Reinforcement Learning for Diffusion Models: On the Importance of Likelihood Estimation Beyond Loss Design"
_ICML.cc/2026/Conference — ICML 2026 regular_

### Official Review · Reviewer_5xBv · 2026-02-23

**Soundness:** 3
**Presentation:** 3
**Significance:** 3
**Originality:** 3
**Overall Recommendation:** 4
**Confidence:** 4

**Summary:**

This paper systematically analyzes the design space of reinforcement learning for fine-tuning diffusion models, challenging the prevailing focus on heavily engineered policy-gradient loss designs inherited from large language models. By disentangling the independent effects of loss objectives, likelihood estimators, and sampling strategies, the authors evaluate both existing techniques and newly proposed exact policy gradient formulations. The study empirically demonstrates that employing an Evidence Lower Bound (ELBO)-based likelihood estimator is the dominant driver of training efficiency and performance, significantly outweighing the impact of the specific policy-gradient objective chosen.

**Compliance With Llm Reviewing Policy:**

Affirmed.

**Final Justification:**

The rebuttal addressed my concerns about the statistical rigor. It looks like the task has a rather small variance, thus the experimental results could serve as evidence to support the conclusion in the paper.

**Key Questions For Authors:**

The major questions:
1. Could the authors clarify how many independent runs were used to generate the point estimates in Table 1, and provide rigorous statistical evaluations, such as bootstrap confidence intervals, to verify that the performance gains are statistically significant rather than an artifact of random seeding?

Some other minor questions:
1. As the ELBO is a biased estimator of likelihood, it is interesting that the ELBO-based likelihood estimation outperforms trajectory-based likelihood estimation. Could the authors explain why? I understand the theoritical unbiased property might not ideal in engineering practice, but I would like to see the deeper reason why, and possibly some ablation studies to verify it.

In general, the paper studied an interesting question. If the authors could fix the major flaw in empirical evaluations, I would suggest acceptance.

---

Update after the rebuttal: All questions have been addressed. I have updated my score.

**Limitations:**

Yes

**Strengths And Weaknesses:**

## Strength

1. The presentation is clear.
2. Multiple likelihood estimation and loss function designs are compared.


## Weakness

1. Some of the background knowledge has factual errors. For example, the naive REINFORCE does not have the KL divergence regularization term in (2).
2. The experimental results are not rigorous. It seems like only one run is conducted for each algorithm. The authors did not mention whether the number is a mean or median statistic over multiple runs, or the confidence intervals over multiple runs. Given that all the results are quite similar, it is difficult to support the claimed results using the existing results without confidence intervals.

---

> ### Author Rebuttal · Authors · 2026-03-31
>
> We thank the reviewer for the constructive feedback. We address each point below.
>
> **W1. REINFORCE naming inaccuracy**
>
> We agree that calling Eq. (2) simply "REINFORCE" is imprecise, since the objective includes a KL regularization term. This is a presentation issue; the method and theoretical results (Theorem 3.1) are unaffected. In the revision, we will rename it to "KL-regularized REINFORCE."
>
> **W2. Statistical rigor / Q1. Number of runs and statistical evaluation**
>
> We agree that multi-seed evaluation strengthens the empirical claims. Our original single-seed reporting followed the evaluation protocol of FlowGRPO, DiffusionNFT, and AWM, which also report single-seed results. During the rebuttal period, we conducted 3 independent runs for PEPG and PAR in the ELBO+ODE setting and report mean ± std below:
>
> | Loss | GenEval | PickScore | ClipScore | HPSv2.1 | Aesthetic |
> |---|---|---|---|---|---|
> | (PEPG) | $0.95 \pm 0.01$ | $22.55 \pm 0.35$ | $0.300 \pm 0.004$ | $0.269 \pm 0.018$ | $5.38 \pm 0.05$ |
> | (PAR) | $0.96 \pm 0.00$ | $22.71 \pm 0.24$ | $0.302 \pm 0.001$ | $0.263 \pm 0.009$ | $5.35 \pm 0.06$ |
>
> Both methods show very low variance across seeds, and the confidence intervals fully overlap, which directly supports our main claim: different loss formulations achieve comparable performance, and the precise choice of loss is not the dominant factor.
>
> | Loss | PEPG (ELBO, ODE) | PAR (ELBO, ODE) | EPG (Traj.) |
> |---|---|---|---|
> | GPU Hours |  $90.0 \pm 4.5$  |  $101.7 \pm 9.6$  |  $750.3 \pm 69.2$ |
> | GenEval | $0.95$ | $0.95$ | $0.93$ |
>
> Regarding the efficiency comparison (ELBO+ODE vs. trajectory+SDE), we report the GPU hours to reach GenEval 0.95 across 3 independent runs: ELBO+ODE reaches this target around 100 GPU hours, compared to 750 for EPG (Traj.) and 423 for FlowGRPO, both of which uses trajectory estimator and SDE sampling. This gap is far too large to be attributed to seed variance and represents a qualitatively different regime of computational cost.
>
>
>
> **Q2. Why ELBO-based estimation enables more efficient optimization**
>
> We would like to first clarify that the trajectory-based estimator is not theoretically unbiased. Both estimators originate from the same Jensen's inequality applied to the log-marginal likelihood. Letting $q(x_{1:T}|x_0)$ denote the fixed forward noising process:
>
> $$\log p\_\theta(x_0) = \log \mathbb{E}\_{q(x_{1:T}|x_0)}\left[\frac{p\_\theta(x_{0:T})}{q(x_{1:T}|x_0)}\right] \geq \mathbb{E}\_{q(x_{1:T}|x_0)}\left[\log \frac{p\_\theta(x_{0:T})}{q(x_{1:T}|x_0)}\right]$$
>
> The right-hand side is the variational lower bound. The ELBO and trajectory estimator are two different ways of evaluating this same lower bound: the former in continuous time via a velocity regression loss, the latter by summing discrete transition log-probabilities. Thus the trajectory estimator is at least as biased as the ELBO, and the relevant question is not "why does biased beat unbiased" but "which biased estimator leads to more efficient optimization in practice."
>
> In this regard, ELBO has a key practical advantage: it decouples likelihood estimation from the sampling process, enabling two degrees of freedom that trajectory-based estimation lacks.
> - First, since ELBO depends only on the final sample $x_0$, any black-box sampler can be used, including efficient ODE solvers with as few as 10 NFEs, whereas the trajectory estimator is inherently coupled to SDE sampling (~40 NFEs).
> - Second, ELBO admits different weighting functions $w(t)$ (path-KL, simple, adaptive; see Sec. 3.2 and Appendix C.1), providing an additional design axis for stable optimization that is unavailable to the trajectory estimator whose weighting is fixed by the noise schedule.
>
> These advantages are verified by existing ablations in the paper. Table 1 (EPG rows) provides a full design-space comparison: trajectory+SDE (0.92) vs. ELBO+SDE (0.90) vs. ELBO+ODE (0.96), demonstrating that ELBO's advantage comes from enabling ODE sampling rather than from the estimator alone. Fig. 4 shows the resulting efficiency gap: ELBO+ODE reaches GenEval 0.95 in ~90 GPU hours vs. ~423 for FlowGRPO (which uses SDE and trajetory estimator), a 4.68× improvement. Fig. 5 further ablates across six ELBO weighting variants, showing that four out of six achieve stable training and strong performance, confirming the robustness of ELBO-based estimation across different weighting choices.

---

> > ### Author Rebuttal · Reviewer_5xBv · 2026-03-31
> >
> > Thank the author for the clarification. All my concerns have been addressed.
> >
> > Regarding the likelihood estimation, I mixed it up with the unbiased policy gradient estimator used in DPPO [1] and also discussed in Eq. 9 in [2]. I apologize for the misunderstanding. I suggest that the authors add discussions about these methods.
> >
> > [1] Ren, Allen Z., et al. "Diffusion policy policy optimization." arXiv preprint arXiv:2409.00588 (2024).
> > [2] Psenka, Michael, et al. "Learning a diffusion model policy from rewards via q-score matching." arXiv preprint arXiv:2312.11752 (2023).

---

> > > ### Author Response · Authors · 2026-04-07
> > >
> > > We sincerely thank the reviewer for the thoughtful clarification and for revisiting the likelihood estimation discussion so carefully. We are very grateful that our rebuttal helped clarify the point, and we greatly appreciate the reviewer’s time and consideration.

---

### Official Review · Reviewer_Tu4N · 2026-03-12

**Soundness:** 2
**Presentation:** 3
**Significance:** 3
**Originality:** 3
**Overall Recommendation:** 4
**Confidence:** 3

**Summary:**

This paper proposes a framework to isolate and evaluate the differents components of policy gradient methods when applied to diffusion models. It includes objectives, likelihood estimations techniques and sampling methods. The paper restates clearly all of these components, and wrap up on the impact they have in terms of efficiency and quality of the trained models.

On the experimental side, the framework is tested on a variety of instances. It compares across the different setups, costs, losses and likelihood estimations techniques through several experiments.

**Compliance With Llm Reviewing Policy:**

Affirmed.

**Final Justification:**

My overall opinion of the paper was already quite positive, and while some limitations on the breadth of validation remain, the rebuttal addresses my main concerns. I therefore maintain my score, whose description still matches my overall assessment well.

**Key Questions For Authors:**

I have two core questions, and one additional minor question.

1. Correct me if I am wrong, but to me, the corectness problem lies in the PAR proof. In PAR, the objective contains the squared term $\frac{1}{2}\left(A(x)-\log \tfrac{\pi_\theta(x)}{\pi_{\theta^{\mathrm{old}}}(x)}\right)^2$. Hence, let's denotes $r(\pi_\theta):=A(x)-\log \tfrac{\pi_\theta(x)}{\pi_{\theta^{\mathrm{old}}}(x)}$. Then $\frac{\partial}{\partial \pi_\theta(x)}\frac12 r(\pi_\theta)^2= r(\pi_\theta)\frac{\partial r(\pi_\theta)}{\partial \pi_\theta(x)}= -\frac{1}{\pi_\theta(x)}\Bigl(A(x)-\log \tfrac{\pi_\theta(x)}{\pi_{\theta^{\mathrm{old}}}(x)}\Bigr)$. Therefore, the first-order condition must contain the extra residual factor $\Bigl(A(x)-\log \tfrac{\pi_\theta(x)}{\pi_{\theta^{\mathrm{old}}}(x)}\Bigr)$, coming from the chain rule on the square. So this step cannot be identical to the PEPG case, where the residual is linear rather than squared. If the appendix writes the same variation as for PEPG, then the PAR derivation is incorrect.

2. Following the main weakness I pointed out, can the empirical claims be validated on at least an other model than SD3.5-Medium (and ideally across different benchmarks). I have noticed table 5 in the appendix, but these results are not analysed or commented. I would need more intuition on these ones, as well as a clear analysis by the authors.

3. The author mentions in line 28 that policy gradient works really well for autoregressive LLMs (that acts in discrete spaces). Do you think that the key finding of the paper transfers to finetuning diffusion language models (dLLMs), that acts on discrete spaces, but (like continuous diffusion models) also suffers from the impossibility of exact likelihood computation, contrary to autoregressive LLMs?

Despite these questions, I think that the paper is good overall, and I really like the fact that it has a key and clear message. So I am slightly in favor for acceptance, and I am open to augment my score further if my questions are solved.

**Limitations:**

A discussion about limitation is not provided. I suggest that the paper should states that the key finding is valid for their particular choice of model and benchmark.

**Strengths And Weaknesses:**

- Strenghts ;

  - The paper treats an understudied problem in the area of post training for diffusion models.
  - The paper comes up with one key finding/actionnable result : likelihood estimation seems to be the main factor, while the loss design should matter less.
  - The paper is really well written, and it's quite clear to understand the background, the existing methods, and the problem it aims to tackle.
  - Empirical results are interesting ; in particular, in the considered experimental setup, the main claim of the paper is well supported empirically through extensive experiments (both qualitatively and quantitatively).

- Weaknesses ;

  - The main weakness I can see is that, despite empricial strong results, they are mainly supported by experiments on one particular model (SD3.5-Medium), as well as one particular family of benchmark. Ablation on at least differents models (and maybe differents benchmarks) seem necessary to fully validate the key finding.
  - I think that there is one correctness issue in the proof of Theorem 3.1 (see questions below).

---

> ### Author Rebuttal · Authors · 2026-03-31
>
> We thank the reviewer for the positive assessment and constructive questions. We address each point below.
>
> **W1,Q2. Validation on other models**
>
> We thank the reviewer for this important suggestion. To provide evidence that our method is not specific to SD3.5-M, we additionally evaluate on FLUX.1-Dev and report results on GenEval with PEPG together with other evaluation metrics. We observe a similar pattern to SD3.5-M: (1) our method substantially improves GenEval and most evaluation rewards, and (2) it reaches similar performance in a comparable number of epochs (about 400 for FLUX.1-Dev versus about 360 for SD3.5-M). We believe this provides supportive evidence that the benefit of our approach is not limited to a single backbone and can transfer to another strong pretrained diffusion/flow model.
>
> |Model|Method|GenEval|PickScore|ClipScore|HPSv2.1|ImgRwd|
> |-|-|-|-|-|-|-|
> |FLUX.1-Dev (w/ cfg)|Baseline|0.66|22.84|0.295|0.274|0.96|
> |FLUX.1-Dev| Ours (ELBO, ODE)|0.93|23.29|0.304|0.279|1.18|
>
> Regarding the generality of our key findings, we believe our key findings about ELBO-based estimation are not specific to a particular model. Its main advantages come from the estimator itself: it avoids chaining $N$ transition probabilities (line 209, eq. (Trajectory)), and is decoupled from the sampler choice. For these reasons, we expect the same advantages to hold across diffusion/flow models with a pretrained velocity field.
>
> That said, we agree that fuller validation would strengthen this claim further. In particular, repeating the same trajectory-vs.-ELBO ablation on FLUX and extending the study to larger-scale or more diverse generative settings remain important future directions, and we will add this as an explicit limitation in the revision.
>
> **W2, Q2. Correctness of PAR derivation**
>
> We thank the reviewer for the careful examination. There is indeed a typo in the PAR objective: both Eq. (PAR) in the main text and Eq. (10) in the appendix are missing a negative sign before the squared term. The correct PAR objective should read: $$L\_{\text{par}}(\theta)=\mathbb{E}_{x^{i}_0\sim\pi\_{\theta\_{old}}}[-\frac{1}{2} \frac{\text{sg}(\pi\_\theta)}{\pi\_{\theta\_\text{old}}}(x_0^i) \| A\_{\text{par}}^i-\log\rho\_\theta(x_0^i)\|^2-\eta KL(x_0^i)]$$ With this correction, the derivation in App. B (Eq. 12) is correct. We walk through the key steps to address the reviewer's concern:
> - **Expand the expectation.** Writing $L$ as an integral, the sampling density $\pi_{\theta_{old}}$ in the expectation cancels with the $\pi_{\theta_{old}}$ in the denominator, giving:$$L=\int-\frac{1}{2}\text{sg}(\pi\_\theta)(x_0)\| A-\log\rho\_\theta(x_0)\|^2dx_0-\eta KL(\pi\_\theta\|\pi\_\text{ref})$$
> - **Take the first variation $\delta L/\delta\pi_\theta$.** The $\text{sg}(\pi_\theta)$ is treated as a constant, so the derivative acts only on $\log\rho_\theta=\log(\pi_\theta /\pi_{\theta_{old}})$ inside the squared norm and on the KL term. The squared term contributes:$$+\text{sg}(\pi_\theta)(x_0) \cdot \frac{1}{\pi_\theta(x_0)} \cdot \left(A - \log\rho_\theta\right)$$
> - **Key simplification.** Since $\text{sg}(\pi_\theta)$ and $\pi_\theta$ have the same value (sg only blocks the gradient), $\text{sg}(\pi_\theta)/\pi_\theta=1$, and we obtain:$$\frac{\delta L}{\delta\pi_\theta}=(A-\log\rho_\theta)-\eta\log\frac{\pi_\theta}{\pi_ref}-\eta+C=0$$ This is exactly Eq. (12), identical to the PEPG first-order condition. The crucial point is that $\text{sg}(\cdot)$ prevents the chain rule from producing the extra residual factor $(A - \log\rho_\theta)$. Without $\text{sg}$, the reviewer's calculation would be correct; with $\text{sg}$, only the inner $\log\rho_\theta$ participates in the variation, yielding a linear first-order condition. Thm 3.1 is therefore unaffected. We will fix the sign typo in the revision.
>
> **Q3. Transfer to diffusion LLMs**
>
> We believe the key finding that likelihood estimation quality dominates loss functional choice transfers to dLLMs. The core reasoning is the same: dLLMs have intractable likelihoods, so any policy-gradient method must rely on approximate likelihood estimation. When the estimation error dominates the differences between loss formulations, the estimator naturally becomes the primary factor.
>
> Notably, recent dLLM RL works such as [1,2] have identified log-probability estimation as a critical bottleneck and devote significant effort to improving it. This independently confirms that likelihood estimation is a central challenge whenever exact likelihoods are unavailable, which is precisely the condition under which our finding holds. Extending our systematic study to dLLMs, disentangling the relative importance of estimation, loss, and sampling in the discrete setting, is a promising direction for future work.
>
> **References**
>
> [1] d1: Scaling Reasoning in Diffusion Large Language Models via Reinforcement Learning
>
> [2] DiffuCoder: Understanding and Improving Masked Diffusion Models for Code Generation

---

> > ### Author Rebuttal · Reviewer_Tu4N · 2026-04-03
> >
> > Thank you to the authors for the careful and constructive rebuttal.
> >
> > The rebuttal satisfactorily addresses my correctness concern regarding the PAR derivation. In particular, after expanding the expectation, the $\pi_{\theta_{\text{old}}}$ term cancels, and the first variation of the corrected PAR objective indeed simplifies to the same linear stationarity condition as in the PEPG case. I therefore consider this point resolved, assuming the sign typo and the derivation are fixed clearly in the revision.
> >
> > On the experimental side, the additional FLUX.1-Dev result is a useful strengthening of the empirical case, and it increases my confidence that the main finding is not purely specific to SD3.5-Medium. That said, I still think the paper’s main claim should be phrased with some care, since the evidence remains limited to a relatively small number of backbones and benchmarks. I appreciate that the authors now acknowledge this explicitly as a limitation.
> >
> > Overall, I view the main contribution primarily as a clear and well-executed empirical disentangling of objective design versus likelihood estimation, rather than as a broadly established universal principle, but I find the message valuable and the paper solid. I thank the authors again for their answers. My overall opinion of the paper was already quite positive, and while some limitations on the breadth of validation remain, the rebuttal addresses my main concerns. I therefore maintain my score, whose description still matches my overall assessment well.

---

> > > ### Author Response · Authors · 2026-04-07
> > >
> > > We thank the reviewer for the careful reading, thoughtful feedback, and positive assessment of our work. We are especially grateful that the rebuttal helped resolve the concern regarding the PAR derivation. We also appreciate the reviewer’s balanced perspective on the scope of our empirical claims. We agree that the current evidence should be presented with appropriate care, and we will make sure the revision clearly reflects both the main message of the paper and its current limitations in validation breadth.

---

### Official Review · Reviewer_dnN8 · 2026-03-13

**Soundness:** 4
**Presentation:** 4
**Significance:** 2
**Originality:** 2
**Overall Recommendation:** 4
**Confidence:** 3

**Summary:**

This work separates three different elements -- sampling, log likihood, and algorithm. It then introduces a family of new algorithms for reinforcement learning based on TRPO and policy gradient. The paper demonstrates good performance of their main algorithm PEPG in conditional image genrieaiotn asks with more better sample efficiency.

**Compliance With Llm Reviewing Policy:**

Affirmed.

**Final Justification:**

The rebuttal successfully addressed my concerns. I believe that presentation of the work can be changed, and made more standard according to RL literature (ie. PEPG should not say it is the TRPO objective, but instead adding an extra KL regularization to it). Otherwise, this is a good paper.

**Key Questions For Authors:**

1.  The first method presented (epg), is an importance weighted version of AWR [0]. Further, it seems that the only difference from AWM is weighting. But AWM performance is much lower than EPG. Why is this the case? (the ablation shows that simple weighting works well too)
2. Can you clarify the derivation for PEPG.
3. KL is with respect to two polices. How is this computed with respect to $KL(x_0)$?

[0] https://arxiv.org/abs/1910.00177

**Limitations:**

yes

**Strengths And Weaknesses:**

**Strengths**
- This paper is well written and well presented.
- Their method, called EPG, is quite effective.
- I enjoyed the perspective of trying to separate out the three components -- likelihood estimation, rollout, and loss function.\
- PEPG algorithm is a novel contribution.

**Weaknesses**
- The ELBO loss presented, would benefit from being presented as in coordination with the flow matching loss (which it is essentially a weighted flow matching loss)
- PEPG seems to not be strongly discussed, regarding its derivation, despite being the primary algorithm in this work. Indeed, I believe that it is from maximizing reward, subject to both a KL penalty and a proximal penalty, but I cannot find this written anywhere. The original TRPO work was written with respect to solely a proximal penalty.

Nit: "and in the sequel," reads weirdly in section 2. Not a big issue.

---

> ### Author Rebuttal · Authors · 2026-03-31
>
> We thank the reviewer for the detailed and constructive feedback.
>
> **General response on significance and originality.** Our contribution is a systematic disentanglement of the key design factors in RL-based diffusion model fine-tuning: policy-gradient objectives, likelihood estimation, and sampling strategy. Through controlled experiments, we show that ELBO-based likelihood estimation is the dominant factor driving efficient optimization, while heavily engineered loss designs (clipping, advantage normalization, CFG) contribute comparatively little. We further show that ODE sampling complements ELBO by exploiting its decoupling from the sampling process, yielding additional efficiency gains. This is a non-obvious finding as the concurrent literature has focused almost entirely on loss engineering, and our result redirects this effort toward what actually matters. Practically, this insight simplifies the design space: practitioners can adopt simple losses like EPG with ELBO + ODE and match or exceed complex GRPO variants, achieving a 4.6× efficiency gain over FlowGRPO. We also contribute three new theoretically grounded objectives (EPG, PEPG, PAR) with convergence guarantees (Thm. 3.1), which serve both as standalone algorithms and as controlled experimental tools. We believe such a systematic design-space study is highly beneficial for the diffusion model post-training community, much as Karras et al. (2022), has been impactful for diffusion model pre-training by identifying the true drivers of performance and simplifying future development.
>
> **W1. ELBO as weighted flow matching**
>
> We agree and thank the reviewer for the suggestion. The standard flow matching loss [1,2] is a special case of the ELBO with simple weighting $w(t)=1$ (Eq. 14). We will add a remark in Sec. 3.2 to make this connection explicit in the revision.
>
> **W2, Q2. PEPG derivation unclear**
>
> The reviewer's understanding is correct. PEPG arises from maximizing expected reward subject to both a proximal penalty to the current policy $\pi_k$ and a KL penalty to the reference policy $\pi_{ref}$:
>
> $$\pi_{k+1}=\arg\max_\pi \mathbb{E}_\pi[R(x_0)]-\frac{\beta}{\eta}KL(\pi\|\pi_k) - \beta KL(\pi\|\pi\_{ref})$$
>
> This admits the closed-form solution (Eq. 8 in App. B):$$\pi_{k+1}\propto\left(\exp(R/\beta)\pi_{ref}\right)^{\frac{\eta}{1+\eta}}\pi_k^{\frac{1}{1+\eta}}$$which is a geometric interpolation between the reward-tilted reference policy and the current policy. As $\eta\to\infty$, the proximal term vanishes and we recover the standard KL-regularized solution $\pi^* \propto\pi_{ref} \cdot \exp(R/\beta)$, which is the EPG target. As $k\to\infty$, the iterates converge to this same target regardless of $\eta$.
>
> In the revision, we will add the optimization problem above to Sec. 3.3 before the formal PEPG objective. We thank the reviewer for highlighting this gap.
>
> **Q1. EPG vs. AWR/AWM performance gap**
>
> We respectfully clarify that EPG is not an importance-weighted version of AWR. The two have fundamentally different motivations. EPG is an on-policy policy gradient objective: its underlying target is $\mathbb{E}\_{\pi_\theta}[A(x)]$, and the importance ratio $\rho\_\theta=\pi_\theta/\pi_{\theta_\text{old}}$ serves to correct for off-policy sampling from $\pi_{\theta_\text{old}}$ (see App. B, Eq. 4). AWR, in contrast, is a purely off-policy weighted regression objective: $\mathbb{E}\_{\pi_\text{old}}[\exp(A/\beta) \cdot \log \pi_\theta(x)]$, where the exponential advantage weights are fixed and independent of the current policy $\pi_\theta$, with no distribution correction. Thus the two differ in both formulation (linear advantage + importance ratio vs. exponential advantage, no importance ratio) and optimization principle (on-policy with correction vs. off-policy regression).
>
> Regarding the performance gap with AWM: we reproduced AWM results using their official codebase. We note that while the AWM paper states CFG is not used, the default codebase implementation includes CFG, which may result in suboptimal hyperparameters for the CFG-free setting we report. Beyond this, several design differences contribute to the gap: AWM uses GRPO with clipping rather than EPG, uses simple rather than adaptive ELBO weighting, does not use EMA for old policy updates, and uses different ODE steps and other hyperparameters. As the reviewer correctly notes, ELBO weighting alone has limited impact (Fig. 5), so the gap is primarily attributable to the combination of the other factors.
>
> **Q3. KL computation between policies**
>
> The per-sample KL is computed as a velocity-matching loss (App. C.3, Eq. 20): $KL(\pi_\theta\|\pi\_{ref}) = \mathbb{E}\_t \mathbb{E}\_{x_t}[w(t)\|v_\theta(x_t, t)-v_{ref}(x_t, t) \|^2 ]$, following the Girsanov-based estimator used in DiffusionNFT and FlowGRPO, with $w(t) = 1$ in practice.
>
> **References**
>
> [1] Flow matching for generative modeling
>
> [2] Flow straight and fast: Learning to generate and transfer data with rectified flow

---

> > ### Author Rebuttal · Reviewer_dnN8 · 2026-04-03
> >
> > I thank the authors for their rebuttal.
> >
> > The discussion has revolved my concerns and I will update my score accordingly.

---

> > > ### Author Response · Authors · 2026-04-07
> > >
> > > We sincerely thank the reviewer for the constructive discussion. We are glad that our response helped resolve the reviewer’s concerns, and we greatly appreciate the careful consideration of our paper.

---

### Official Review · Reviewer_J9uv · 2026-03-13

**Soundness:** 3
**Presentation:** 3
**Significance:** 3
**Originality:** 3
**Overall Recommendation:** 5
**Confidence:** 3

**Summary:**

RL fine-tuning for diffusion-based text-to-image models has recently become popular, with different methods making different choices about the policy-gradient objective, likelihood surrogate, and sampling procedure. This paper systematically studies these design choices and asks which of them actually drives training efficiency and final performance. Through controlled comparisons, the authors argue that the dominant factor is the likelihood estimator, and in particular that ELBO-based likelihood estimation is substantially more effective and efficient than trajectory-based alternatives, while the exact policy-gradient loss matters less. Based on this analysis, they combine ELBO-based likelihood estimation with efficient sampling to form a simple training recipe that achieves state-of-the-art or near-state-of-the-art results on benchmarks such as GenEval, with improved training efficiency.

**Compliance With Llm Reviewing Policy:**

Affirmed.

**Key Questions For Authors:**

- Many of the motivations in Sec. 3.1 are imported from the LLM RL literature. How well do these arguments transfer to diffusion models, and are they specific to text-to-image settings?
- By removing advantage normalization, does the method introduce cross batch scale instability during GRPO style training?

**Limitations:**

yes

**Strengths And Weaknesses:**

**Strengths**
-  The paper provides a useful design-space study for RL fine-tuning of diffusion models. This is a valuable perspective for understanding which components matter most in practice.
- Empirically, the proposed method appears to improve the efficiency of policy gradient based finetuning(FlowGRPO) in this setting by using the ELBO based likelihood estimation. This method also remaining competitive with strong baselines such as DiffusionNFT.
- The experimental presentation is clear. The plots and tables are generally easy to follow.

**Weakness**
- The paper does not include a generalization evaluation comparable to the one reported in FlowGRPO (Tab 4) There is also no diversity comparison between different methods.
- Section 3.2 needs citations for forward/backard ELBO formula.
-  Some of the simplified design choices are not sufficiently justified. For example, the paper argues that components such as clipping can be removed, but the explanation for why these components are unnecessary is currently limited

---

> ### Author Rebuttal · Authors · 2026-03-31
>
> We thank the reviewer for the positive assessment and constructive suggestions.
>
> **W1. Generalization and diversity evaluation**
>
> **Generalization.** We agree and have added evaluations following FlowGRPO Tab. 4. We evaluate generalization on unseen object classes and unseen counting scenarios from GenEval:
>
> |Method|Overall|Single|Two|Count|Color|Position|
> |-|-|-|-|-|-|-|
> |SD3.5-M|0.64|0.96|0.73|0.53|0.87|0.26|
> |+Flow-GRPO|0.90|1.00|0.94|0.86|0.97|0.84|
> |+Ours (ELBO, ODE)|0.93|1.00|0.96|0.91|0.99|0.91|
>
> We also note that DrawBench, which uses prompts entirely disjoint from training, already serves as a generalization test in Tab. 2, where our method achieves competitive results.
>
> **W2. Missing ELBO citations**
>
> We thank the reviewer for pointing this out. In the revised manuscript, we will add citations for the forward ELBO formula [1,2,3] and the backward trajectory-based formula [4] in Sec. 3.2. These derivations are already discussed with full references in Appendix C.1; the omission in the main text was an oversight.
>
> **W3. Justification for removing clipping, normalization, CFG**
>
> Our approach is based on policy-gradient loss functionals that are *exact*, meaning that they are theoretically guaranteed to converge to the target distribution (Thm. 3.1), rather than being motivated primarily by GRPO-style training heuristics. From this perspective, common design choices such as clipping, advantage normalization, and CFG are not essential components of the objective itself. Empirically, we further find that these heuristics are not the main determinants of performance in our setting:
>
> - **Clipping.** GRPO with clipping (ELBO+ODE) achieves GenEval 0.94, whereas EPG, PEPG, and PAR without clipping achieve comparable or slightly better performance (around 0.96; Tab. 1).
> - **Advantage normalization.** Dividing by the within-group standard deviation can introduce prompt-difficulty bias [5], and this effect may be amplified in diffusion RL, where rewards are continuous and often exhibit small variance.
> - **CFG.** As shown in Sec. 4.3, strong performance is achievable without CFG, while also reducing sampling cost by approximately $2\times$. Although the no-CFG setting may begin with lower performance, it improves rapidly and reaches a competitive final result.
>
> Overall, these results suggest that competitive or better performance can be obtained even with simplified objectives and training settings across all loss functionals we study. Therefore, our conclusion is not that these heuristics are always unnecessary, but rather that they are not the key design factor in our setting. Instead, the more important factors are the likelihood estimation method and the sampling strategy.
>
> **Q1. Transferability of LLM RL motivations**
>
> The motivations transfer naturally because both settings share the same core structure: a generative policy is optimized against a reward function using policy-gradient methods, typically with KL regularization to control deviation from a reference policy (Eq. 1). Our work follows this principle: we design new objectives (PEPG, PAR), deploy an existing one (EPG), and also evaluate GRPO, to systematically assess whether the choice of policy-gradient loss is the dominant factor.
>
> That said, the key finding that likelihood estimation matters more than loss design is *specific* to the diffusion setting and does not arise in LLM-RL. Autoregressive LLMs have exact, efficiently computable likelihoods, so likelihood estimation is never a bottleneck. Diffusion models lack this property, making the choice of likelihood estimator a critical and previously underexplored design axis.
>
> Finally, the framework itself is not specific to text-to-image. Alg. 1 only assumes access to a pretrained velocity field and a reward function, so it is broadly applicable to diffusion or flow models. We view extensions to domains such as text-to-video as an important direction for future work.
>
> **Q2. Scale instability without advantage normalization**
>
> In practice, we did not observe cross-batch scale instability. Two mechanisms provide implicit scale control: (1) the KL regularization term naturally penalizes large policy deviations regardless of reward magnitude, and (2) the EMA-based policy update (Alg. 1, line 14) ensures that the reference policy evolves slowly, preventing sudden scale shifts in the importance ratio. The training curves in Figs. 5, 6 confirm stable optimization across all settings. In diffusion RL, reward values within a group are continuous with relatively small variance, so dividing by the standard deviation risks amplifying noise rather than stabilizing training.
>
> **References**
>
> [1] Score-Based Generative Modeling through Stochastic Differential Equations
>
> [2] Variational Diffusion Models
>
> [3] Demystifying Diffusion Objectives: Reweighted Losses are Better Variational Bounds
>
> [4] Denoising Diffusion Probabilistic Models
>
> [5] Understanding r1-zero-like training: A critical perspective

---

> > ### Author Rebuttal · Reviewer_J9uv · 2026-04-04
> >
> > Thanks for answering my questions on the justifications for clipping, CFG and the advantage function. I maintain my rating.

---

> > > ### Author Response · Authors · 2026-04-07
> > >
> > > We thank the reviewer for the thoughtful follow-up and for carefully considering our clarifications regarding clipping, CFG, and the advantage function. We appreciate the reviewer’s time and consideration.

---

### Decision · Program_Chairs · 2026-04-30

**Decision:**

Accept (regular)

**Comment:**

*Motivation:* Analyzing RL methods developed for post-training of diffusion models and identifying the key challenge in the design.


*Contribution:* Decoupling and characterizing the influence of three different factors including policy gradient loss function, likelihood estimation, and sampling in the design pipeline of RL for diffusion. Showing that likelihood estimation is the main factor.

*Reviews summary:* Reviewers believe that this paper has an interesting and significant contribution in shaping the future direction of research by identifying the key factor in employing RL for diffusion models. The paper is well written and motivated based on a good research story.

*Rebuttal updates:* There were concerns about the proof of Theorem 3.2 and also the standard deviation over random runs, which were addressed in the rebuttal.

*Conclusion:* The AC votes for weak acceptance, as the paper provides novel insights well supported by experiments.